# Programmable epigenome editing by transient delivery of CRISPR epigenome editor ribonucleoproteins

Da Xu[1,8], Swen Besselink [1,2,8], Gokul N. Ramadoss [3], Philip H. Dierks [3], Justin P. Lubin [4], Rithu K. Pattali[1], Jinna I. Brim [1], Anna E. Christenson [1], Peter J. Colias[1], Izaiah J. Ornelas[1], Carolyn D. Nguyen[1], Sarah E. Chasins [4,5], Bruce R. Conklin[3,6,7] & James K. Nuñez [1,5] ✉

Programmable epigenome editors modify gene expression in mammalian cells by altering the local chromatin environment at target loci without inducing DNA breaks. However, the large size of CRISPR-based epigenome editors poses a challenge to their broad use in biomedical research and as future therapies. Here, we present Robust ENveloped Delivery of Epigenome-editor Ribonucleoproteins (RENDER) for transiently delivering programmable epigenetic repressors (CRISPRi, DNMT3A-3L-dCas9, CRISPRoff) and activator (TET1-dCas9) as ribonucleoprotein complexes into human cells to modulate gene expression. After rational engineering, we show that RENDER induces durable epigenetic silencing of endogenous genes across various human cell types, including primary T cells. Additionally, we apply RENDER to epigenetically repress endogenous genes in human stem cell-derived neurons, including the reduction of the neurodegenerative disease associated V337M-mutated Tau protein. Together, our RENDER platform advances the delivery of CRISPR-based epigenome editors into human cells, broadening the use of epigenome editing in fundamental research and therapeutic applications.

Advances in genome and epigenome editing have greatly broadened the ability to manipulate human cells, fueling both fundamental research and therapeutic applications[1–3]. Epigenome editing represents an attractive approach for regulating gene expression without inducing potentially harmful DNA breaks or permanently changing the underlying genome sequence[4–14]. Instead, programmable transcriptional repression or activation is induced by altering epigenetic modifications on DNA and histones, typically by epigenome editing tools that consist of epigenetic effector domains derived from transcriptional repressors and activators that are synthetically coupled to programmable DNA-binding proteins such as zinc finger (ZF),

transcription activator-like effector (TALE) or catalytically deactivated Cas (dCas) proteins[15].

Recent developments in epigenome editing include methods to program durable epigenetic repression of endogenous mammalian genes by transient expression of synthetic epigenetic repressors that simultaneously establish de novo DNA methylation and repressive H3K9me3 at target gene promoters[16,17]. Such programmable hit-and-run epigenome editing tools combine the DNMT3A-DNMT3L (DNMT3A-3L) enzymatic complex to catalyze de novo DNA methylation and the Krüppel-associated box (KRAB) transcriptional repressor domain that recruits cofactors to deposit H3K9me3. Recently, ZF- and

---

[1]Department of Molecular and Cell Biology, University of California, Berkeley, CA, USA. [2]Molecular Mechanisms of Disease Master's Program, Radboud University, Nijmegen, the Netherlands. [3]Gladstone Institutes, San Francisco, CA, USA. [4]Department of Electrical Engineering and Computer Sciences, University of California, Berkeley, CA, USA. [5]Chan Zuckerberg Biohub San Francisco, San Francisco, CA, USA. [6]Department of Cellular and Molecular Pharmacology, University of California, San Francisco, CA, USA. [7]Department of Medicine, University of California, San Francisco, CA, USA. [8]These authors contributed equally: Da Xu, Swen Besselink. ✉e-mail: jamesnunez@berkeley.edu

TALE-based hit-and-run epigenome editors have been applied in mouse models to durably repress *Pcsk9* in the liver and a prion gene in the brain[18,19]. While these approaches highlight the potential for in vivo therapeutic epigenome editing, the reliance on ZF and TALE DNA-binding platforms restricts multiplexing and easily programmable capabilities offered by RNA-guided CRISPR-dCas9.

Previously, we engineered the CRISPRoff epigenome editor by synthetically fusing dCas9 to DNMT3A-3L and KRAB epigenetic effector domains for programmable multiplexed durable gene silencing across various cell types and for pooled genome-wide screens[17]. However, the large size of CRISPR-based epigenome editors, including CRISPRoff, poses a significant barrier to their efficient delivery into human cells, thereby limiting their broader use. For example, CRISPRoff (~7.2 kilobase pairs) exceeds the packaging capacity of adeno-associated virus (AAV) delivery vectors, and prolonged AAV expression of epigenome editors may increase the risk of off-target editing[1,20]. Additionally, mRNA-based delivery of CRISPR reagents requires electroporation, which can be potentially cytotoxic, or as lipid nanoparticle (LNP)-mRNA complexes, which require cell type-specific formulations, remaining a current challenge[21–23].

Alternatively, delivering CRISPR reagents as ribonucleoprotein (RNP) complexes offers multiple benefits[21,22,24]. First, RNP delivery does not require transgene or viral vector-based expression, thus there is no potential risk of viral DNA integration. Second, RNP is the most transient delivery formulation of CRISPR editors, thereby minimizing exposure time in target cells and decreasing off-target editing. However, many commonly used epigenetic effectors, such as KRAB and MeCP2, contain intrinsically unstructured regions that are challenging to purify in vitro for RNP delivery methods[25,26]. Thus, there is an unmet need for a platform to transiently deliver CRISPR epigenome editors as RNPs into mammalian cells for modulating gene expression.

Virus-like particles (VLPs), or enveloped delivery vehicles (EDVs), derived from retroviruses such as the murine leukemia virus or human immunodeficiency virus, have emerged recently as efficient vehicles for transient delivery of CRISPR-based genome editors as RNP complexes[27–34]. VLPs provide several unique advantages over other delivery methods for CRISPR technologies. First, VLPs are less limited by cargo size, making them well-suited to deliver large CRISPR cargos. Second, VLPs are composed of viral proteins that encapsulate their cargo in a protective shell and possess the natural ability to enter cells, but lack viral genetic material, thereby eliminating the risk of viral genome integration[35]. As such, VLP-based delivery has been adapted for CRISPR nucleases, base editors, and prime editors, highlighting the diverse cargo payload of VLPs[27–31]. To our knowledge, VLPs have not been explored yet for RNP delivery of CRISPR epigenome editors.

Here, we leverage engineered virus-like particles to transiently deliver various CRISPR epigenome editor RNPs for programmable gene silencing and activation in human cells. We term our platform RENDER, Robust ENveloped Delivery of Epigenome-editor Ribonucleoproteins. We apply RENDER for efficient epigenetic silencing of endogenous genes that persists for weeks and is broadly applicable across various human cell types, including donor-derived primary T cells. We demonstrate that RENDER-based delivery of CRISPRoff into induced pluripotent stem cell-derived neurons results in durable repression of the V337M-mutated *MAPT* gene, thereby reducing protein levels of the neurodegenerative disease associated Tau protein. We envision our RENDER platform for transient delivery of CRISPR epigenome editors to advance the broad use of programmable epigenome editing in biomedical research and therapeutic applications.

## Results

### eVLPs enable delivery of various epigenome editor RNP into human cells

Previously, the murine leukemia virus has been engineered for VLP delivery of Cas9 nucleases, base editors (BE), and prime editors[28,30,31].

We hypothesized that these engineered virus-like particles (eVLPs) could also be adapted to deliver epigenome editors as RNP complexes (Fig. 1a). To test this, we modified the BE-eVLP platform by substituting the base editor in the gag-editor fusion protein with various epigenome editors, including CRISPRi, DNMT3A-3L-dCas9, and CRISPRoff (Fig. 1b)[30]. Furthermore, as several studies have reported improved epigenetic silencing with the ZIM3 KRAB domain compared to the widely-used KOX1 KRAB domain, we aimed to directly compare the two KRAB domains for CRISPRi and CRISPRoff in the eVLP delivery system[36,37]. As control, we used dCas9 with no epigenetic effector domain fused. Epigenome editor eVLPs were produced by co-transfecting Lenti-X HEK293T cells with plasmids encoding the vesicular stomatitis virus G envelope protein VSV-G, wild-type gag-pol polyprotein, gag-epigenome editor fusion protein, and single-guide RNA (sgRNA) (Fig. 1a). We harvested eVLPs at 48- and 72-hours post-transfection and confirmed the presence of editor proteins in the eVLPs using ELISA (Fig. 1c). Quantification of editor proteins indicates that extending the expression period to 72 h substantially increased the yield compared to the previously reported protocol[30]. Western blot analysis of concentrated CRISPRoff-eVLPs further confirmed successful packaging of CRISPRoff proteins in the eVLP as pre- and post-cleavage species from the gag-CRISPRoff fusion protein (Fig. 1d).

To assess whether eVLPs can deliver functional epigenome editors into cells, we co-packaged eVLPs with each epigenome editor and a sgRNA targeting the *CLTA* promoter. Subsequently, actively dividing HEK293T cells that stably express an endogenous GFP-tagged *CLTA* gene were treated with a single dose of the different epigenome editor eVLPs (Fig. 1e)[38]. Successful epigenetic silencing is signified by a reduced CLTA-GFP protein expression, which can be measured quantitatively in single cells using flow cytometry. Three days post-treatment, we observed varying levels of CLTA-GFP silencing by the different epigenome editor eVLPs, with dCas9-eVLPs showing only modest CLTA-GFP repression in 25% of the treated cells (Fig. 1f). Remarkably, eVLPs packaged with CRISPRi, DNMT3A-3L-dCas9 or CRISPRoff induce robust epigenetic silencing, resulting in over 75% of treated cells with CLTA-GFP silenced (Figs. 1f, g, and Supplementary Fig. 1a). Furthermore, the ZIM3 KRAB domain induces higher epigenetic silencing compared to KOX1 within the CRISPRi-eVLPs, whereas both KRAB domains show comparable silencing efficiency within CRISPRoff (Supplementary Fig. 1b, c).

Then, we monitored the durability of epigenetic silencing over 14 days after a single dose of our different epigenome editor eVLPs. Cells treated with dCas9-eVLPs or CRISPRi-eVLPs fully reactivate CLTA-GFP expression to pre-treatment levels after 7 days, consistent with the transient epigenetic silencing activity of CRISPRi (Fig. 1h and Supplementary Fig. 1d). In contrast, cells treated with DNMT3A-3L-dCas9-eVLPs or CRISPRoff-eVLPs maintain robust epigenetic silencing due to the heritability of the programmed DNA methylation at the *CLTA* promoter as we have shown previously[17] (Fig. 1h). Together, these findings demonstrate that the eVLP platform can effectively package and deliver functional epigenome editor RNPs into human cells to induce transient repression (CRISPRi) or durable epigenetic silencing (DNMT3A-3L-dCas9, CRISPRoff). We refer to our eVLP-mediated packaging and delivery of CRISPR epigenome editors as RENDER, Robust ENveloped Delivery of Epigenome-editor Ribonucleoproteins.

### Reversible epigenome editing by RENDER-TET1-dCas9

An advantage of epigenome editing is the reversibility of the programmed epigenetic modifications at target loci[17,39]. We next applied RENDER to deliver the TET1-dCas9 (v4) epigenome editor that can enzymatically remove repressive DNA methylation from target gene promoters to activate gene transcription[17]. We packaged TET1-dCas9 in eVLPs and treated actively dividing HEK293T cells in which CLTA-GFP has been durable repressed by CRISPRoff (Fig. 1i). Compared to untreated cells, we observe increased CLTA-GFP expression in ~6% of

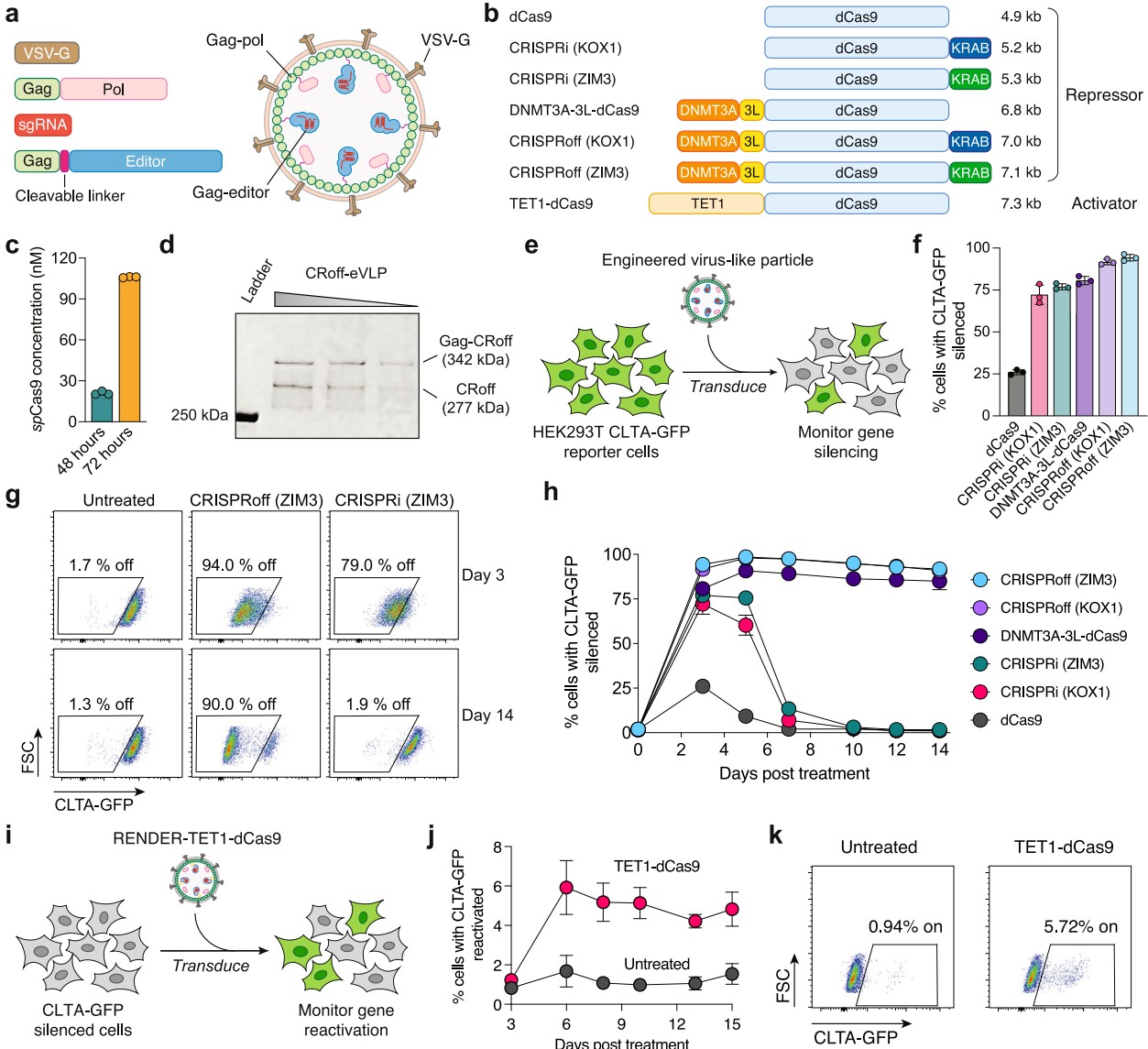

**Fig. 1 | Virus-like particle delivery of epigenome editor RNPs. a** Schematic of a virus-like particle packaging epigenome editors. **b** Schematic overview of different CRISPR-based epigenome editors and their sizes. kb; kilobase pairs. **c** *S. pyogenes* Cas9 (*sp*Cas9) protein quantifications of CRISPRoff-eVLP collected at 48 h and 72 h post-transfection. Protein contents were measured by anti-Cas9 ELISA. Data are shown as individual data points and mean ± SD for n = 3 biological replicates. **d** Western blot evaluating the packaging of CRISPRoff protein in eVLPs at 4, 2 and 1 µl of concentrated CRISPRoff-eVLP with anti-Flag antibody. CRoff; CRISPRoff. **e** Schematic of engineered virus-like particle transduction and CLTA-GFP silencing in HEK293T reporter cells. **f** Quantification of CLTA-GFP silencing in HEK293T cells 3 days post-treatment with different epigenome editor eVLPs (16 µl). **g** Representative flow cytometry plots of CLTA-GFP expression in HEK293T cells at

day 3 and 14 post-treatment with CRISPRi-eVLPs and CRISPRoff-eVLPs (16 µl). **h** Time course of CLTA-GFP silencing in HEK293T cells post-treatment with different epigenome editor eVLPs (16 µl). Values and error bars reflect mean ± SD of n = 3 biological replicates. **i** Schematic of engineered virus-like particle transduction and CLTA-GFP reactivation in HEK293T reporter cells. **j** Time course of CLTA-GFP reactivation in HEK293T cells post-treatment with RENDER-TET1-dCas9 (32 µl). Values and error bars reflect mean ± SD of n = 3 biological replicates. **k** Representative flow cytometry plots of CLTA-GFP expression in HEK293T cells 15 days post-treatment with RENDER-TET1-dCas9 (32 µl). Data are shown as individual data points and mean ± SD for n = 3 biological replicates. Source data are provided as a Source Data file.

cells at 6 days post-treatment that remains stable for 15 days (Fig. 1j and k). We envision that further engineering will improve the reactivation efficiency of RENDER-TET1-dCas9, including co-packaging with transcriptional activator domains to recruit the transcriptional machinery, as we have shown previously with CRISPRon[17].

### Rational optimization of RENDER-CRISPRoff

An advantage of virus-like particle delivery platforms is the ability to perform dose-dependent treatments to titrate the amount of editors delivered into cells. Indeed, varying the dose with our RENDER platform results in titratable degrees of CLTA-GFP silencing in

HEK293T cells for each epigenome editor repressor we tested (Figs. 2a, b and Supplementary Fig. 2a–d). Noticeably, we observed that lower doses of RENDER-CRISPRoff result in weaker durability of gene silencing over a 14-day time course (Fig. 2a). Given that the eVLP delivery platform was optimized initially for CRISPR base editors that differ in size and biochemical properties from epigenome editors, we sought to refine the RENDER platform tailored specifically for CRISPRoff (ZIM3) (Fig. 2c).

First, we hypothesized that optimizing the cargo-to-vector ratio will increase the amount of CRISPRoff RNPs within the eVLPs, thereby improving the efficacy of epigenetic silencing. While keeping the

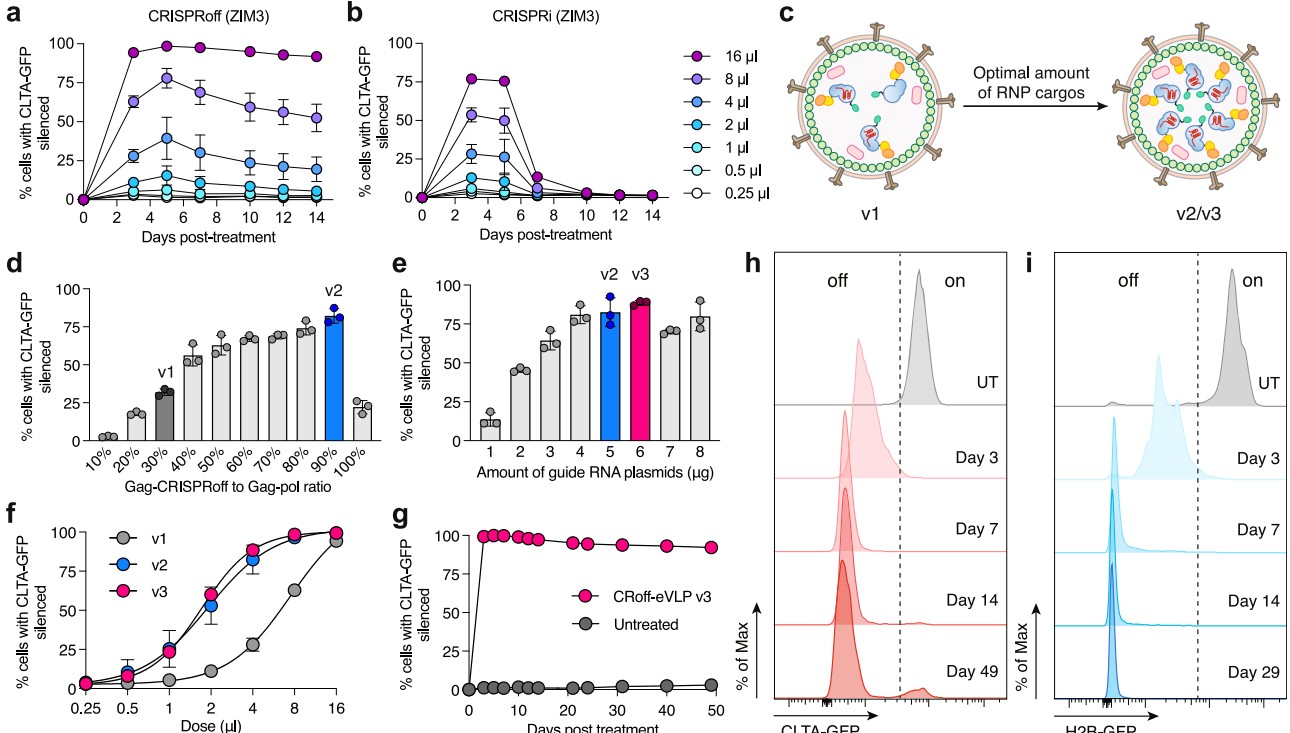

**Fig. 2 | Optimizing RENDER-CRISPRoff composition. a** Time course of CLTA-GFP silencing in HEK293T cells post-treatment with different doses of RENDER-CRISPRoff (ZIM3) (v1). **b** Time course of CLTA-GFP silencing in HEK293T cells post-treatment with different doses of RENDER-CRISPRi (ZIM3). In **a**, **b**, Values and error bars reflect mean ± SD of n = 3 biological replicates. **c**, Schematic of optimization of RENDER-CRISPRoff. **d** Quantification of CLTA-GFP silencing in HEK293T cells 5 days post-treatment with 4 µl RENDER-CRISPRoff with varying gag-CRISPRoff to gag-pol plasmid ratios. X-axis indicates % gag-CRISPRoff plasmid of total amount of gag-CRISPRoff and gag-pol plasmids. Dark gray indicates v1, blue indicates v2. **e** Quantification of CLTA-GFP silencing in HEK293T cells 5 days post-treatment with 4 µl RENDER-CRISPRoff with varying amounts of sgRNA expressing plasmid. Blue

indicates v2, magenta indicates v3. **f** Dose-response curve of CLTA-GFP silencing in HEK293T cells 5 days post-treatment with RENDER-CRISPRoff v1, v2 and v3. Values and error bars reflect mean ± SD of n = 3 biological replicates. Data were fit to four-parameter logistic curves using nonlinear regression. **g** Time course of CLTA-GFP silencing in HEK293T cells post-treatment with RENDER-CRISPRoff v3 (16 µl). Values and error bars reflect mean ± SD of n = 3 biological replicates. **h** Representative histogram plots of CLTA-GFP expression in HEK293T cells at day 3, 7, 14, and 49 post-treatment with RENDER-CRISPRoff v3 (16 µl). UT; untreated. **i** Representative histogram plots of H2B-GFP expression in HEK293T cells at day 3, 7, 14, and 29 post-treatment with RENDER-CRISPRoff v3 (10 µl). UT; untreated. Source data are provided as a Source Data file.

amounts of other plasmids constant, we varied the ratio of gag-CRISPRoff to gag-pol plasmids during co-transfection to package RENDER-CRISPRoff. Intriguingly, we found that eVLPs can accommodate a high proportion of CRISPRoff, with a 90:10 ratio of gag-CRISPRoff to gag-pol plasmids achieving the most effective gene silencing (Fig. 2d). This formulation—demonstrating 99% silencing at maximum dose and a 3.9-fold improved potency (EC50)—is termed RENDER-CRISPRoff v2 (Supplementary Fig. 3a).

Next, we hypothesized that an increased number of CRISPRoff proteins within the eVLP will necessitate additional sgRNA to form functional RNP complexes. Thus, we titrated the amount of sgRNA plasmids in our plasmid transfections for RENDER-CRISPRoff packaging. Our results show that increasing the amount of sgRNA further improves the editing efficiency at non-saturating doses, yielding RENDER-CRISPRoff v3 (Fig. 2e and Supplementary Fig. 3b). These findings demonstrate that by optimizing the component ratios of CRISPRoff-eVLPs, we improve its potency by 4.4-fold compared to v1, thereby lowering the dose required to achieve efficient target gene silencing (Fig. 2f). A single dose of the optimized RENDER-CRISPRoff v3 induces highly efficient and durable gene silencing, with 92% of cells maintaining stable repression of CLTA-GFP at 49 days post-treatment (Figs. 2g, h). To confirm that the optimized RENDER-CRISPRoff v3 can achieve durable silencing across different target genes, we further evaluated its performance in HEK293T cells expressing an endogenous GFP-tagged *H2B* gene, which resulted in durable silencing in 99% of cells at 29 days post-treatment (Fig. 2i).

## RENDER-CRISPRoff editing across various human cell types

After optimization in HEK293T reporter cells, we aimed to evaluate the broad applicability of RENDER-CRISPRoff across different human cell types. To this end, we tested RENDER-CRISPRoff in eight human cell lines originating from different tissues: embryonic kidney (HEK293T), liver (HepG2), leukemic T cells (Jurkat), retinal epithelium (hTERT RPE-1), glioblastoma (U87 MG), lymphoblast (K562), cervical cancer (HeLa), and neuroblastoma (SH-SY5Y) cells. We packaged CRISPRoff with sgRNAs targeting either the promoter of endogenous gene *CD55* or *CD81*, which encode cell surface proteins that are nonessential for cell viability. Seven days post-treatment, we performed antibody staining for CD55 and CD81 to measure gene silencing in single cells using flow cytometry. RENDER-CRISPRoff v3 induces robust silencing of *CD55* in HEK293T, HepG2, Jurkat, hTERT RPE-1, and U87 MG cells (>80% silenced), whereas *CD81* silencing is efficient in Jurkat, hTERT RPE-1, U87 MG, HeLa, and SH-SY5Y cells (>44% silenced) (Fig. 3a). Notably, we observed weaker silencing efficiency for *CD81* in HEK293T and *CD55* in HeLa, which we hypothesize is not a RENDER limitation, but rather due to differences in gene regulatory landscapes across different cell types as shown previously[40–42]. We also detected overall lower silencing efficiency in K562 cells (~22%). To investigate whether this is due to cell type-specific limitation in virus-like particle delivery, we performed dose titration of RENDER-CRISPRoff v3 treatment in K562 cells. This analysis revealed a dose-dependent increase in silencing efficiency of both *CD55* and *CD81* genes (Supplementary Fig. 4a). Additionally, we found that RENDER-CRISPRoff v3 silencing efficiency in K562 cells can

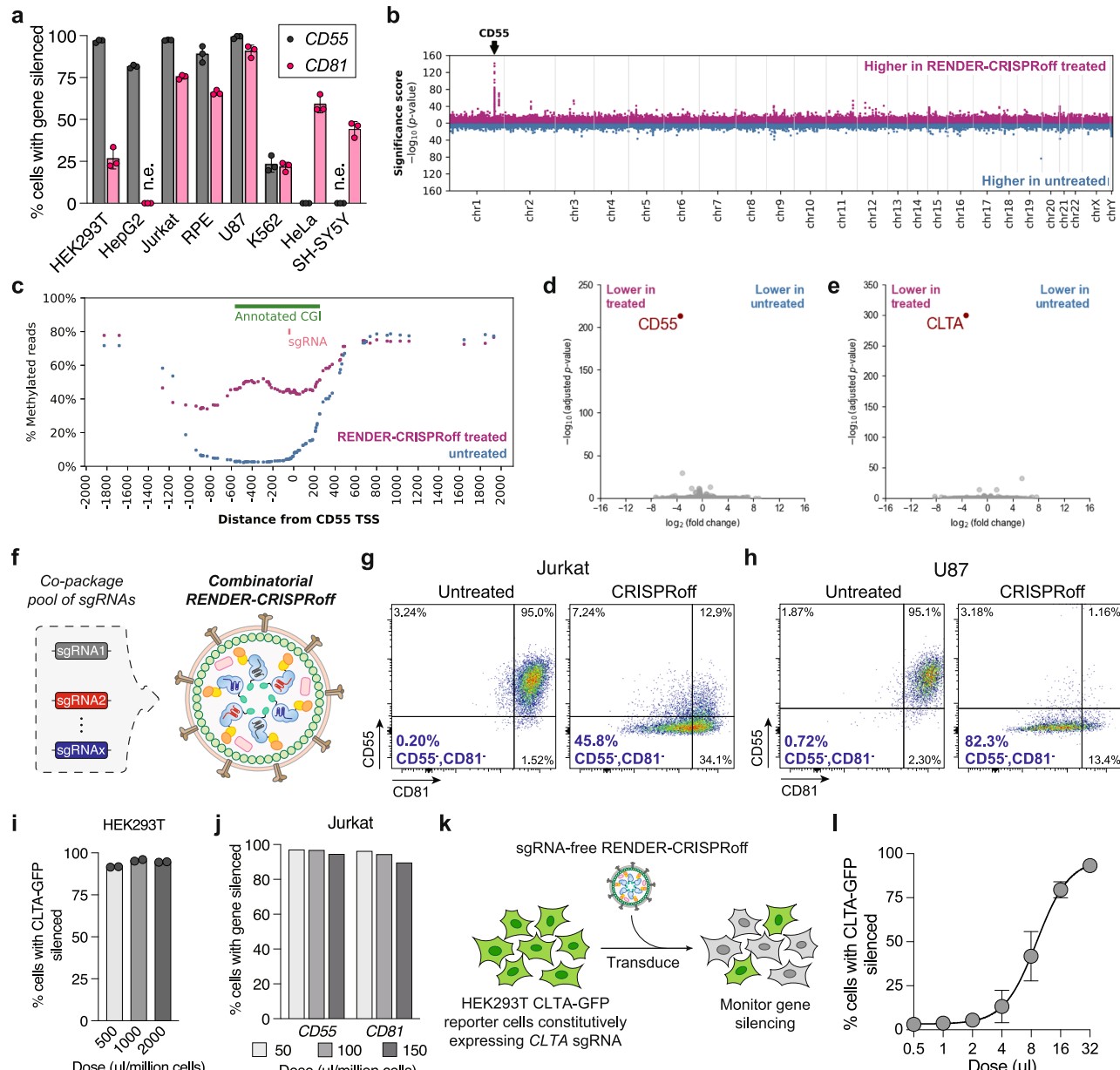

**Fig. 3 | Multiplexed epigenetic silencing across diverse human cell types with RENDER-CRISPRoff. a** Quantification of CD55 and CD81 silencing in different cell lines 7 days post-treatment with RENDER-CRISPRoff v3 (8 µl). Data are shown as individual data points and mean ± SD for n = 3 biological replicates. n.e.; not expressed. **b** Manhattan plot displaying differentially methylated CpGs between Jurkat cells treated with RENDER-CRISPRoff v3 targeting *CD55* and untreated cells (14 days post-treatment) analyzed by WGEM-seq. Magenta dots represent CpGs that are more methylated in RENDER-CRISPRoff-treated cells and blue dots represent CpGs that are more methylated in untreated cells. The arrow denotes the genomic location of the *CD55* gene. **c** Plot displaying the % of WGEM-seq reads that are methylated at each CpG across a ± 2 kilobase pairs window from the *CD55* transcription start site (TSS) between Jurkat cells treated with RENDER-CRISPRoff v3 targeting *CD55* (magenta) or untreated cells (blue) at 14 days posttreatment. Annotated CpG island (CGI) is shown in green and *CD55* targeting sgRNA in red. **d** Volcano plot of RNA-seq data analyses showing differential gene expression between RENDER-CRISPRoff *CD55*-sgRNAs treated Jurkat cells and untreated Jurkat cells (n = 2). The adjusted *p* value is a two-sided *p* value calculated from a DESeq2 Wald test with Benjamini–Hochberg multiple comparisons adjustment. *CD55* is shown as a red dot; all other genes are shown as light gray circles. **e** Volcano plot of RNA-seq data analyses showing differential gene expression between RENDER-CRISPRoff *CLTA*-sgRNAs treated HEK293T cells and untreated HEK293T cells

(n = 2). The adjusted *p* value is a two-sided *p* value calculated from a DESeq2 Wald test with Benjamini–Hochberg multiple comparisons adjustment. *CLTA* is shown as a red dot; all other genes are shown as light gray circles. **f** Schematic of combinatorial sgRNAs packaging into one RENDER virus-like particle. **g** Representative flow cytometry plots of *CD55* and *CD81* expression in Jurkat cells 7 days post-treatment with combinatorial RENDER-CRISPRoff v3 targeting *CD55* and *CD81* (8 µl). **h** Representative flow cytometry plots of *CD55* and *CD81* expression in U-87 MG cells 7 days post-treatment with combinatorial RENDER-CRISPRoff v3 targeting *CD55* and *CD81* (8 µl). **i** Quantification of CLTA-GFP silencing in HEK293T cells 3 days post-treatment of 1.0 × 10⁶ cells with different doses of RENDER-CRISPRoff v3. Data are shown as individual data points and mean ± SD for n = 2 biological replicates. **j** Quantification of *CD55* and *CD81* silencing in Jurkat cells 3 days post-treatment of 5.0 × 10⁵ cells with different doses of RENDER-CRISPRoff v3. Values represent one biological replicate. **k** Schematic of sgRNA-free delivery of CRISPRoff apoprotein by RENDER and CLTA-GFP silencing in HEK293T reporter cells stably expressing a *CLTA* targeting sgRNA. **l** Dose-response curve of CLTA-GFP silencing in HEK293T cells expressing a *CLTA* targeting sgRNA 3 days post-treatment with sgRNA-free RENDER-CRISPRoff. Values and error bars reflect mean ± SD of n = 3 biological replicates. Data were fit to four-parameter logistic curves using nonlinear regression. Source data are provided as a Source Data file.

be further improved using viral transduction enhancers such as polybrene and RetroNectin (Supplementary Fig. 4b, c). These results demonstrate the versatility and efficiency of the RENDER platform to deliver CRISPRoff in diverse human cell types, underscoring its potential for broader applications.

## Specificity profile of RENDER-CRISPRoff treatment

Given that RENDER-CRISPRoff exhibits robust target gene silencing across various cell types, we next profiled its epigenome editing. We first performed genome-wide DNA methylation profiling using enzymatic methyl sequencing (EM-seq) of Jurkat cells 16 days after a single dose of RENDER-CRISPRoff v3 targeting *CD55*. EM-seq analysis showed a significant increase of DNA methylation at the *CD55* gene after RENDER-CRISPRoff v3 treatment as compared to untreated cells (Fig. 3b). Notably, the DNA methylation enrichment spans a ~1.5 kb region centered on the *CD55* promoter, encompassing the annotated CpG island (CGI) and extends ~200 bp upstream of the annotated CGI (Fig. 3c and Supplementary Fig. 5), consistent with our previous observation using CRISPRoff expressed by plasmid transfection[17]. We further analyzed the top ten differentially methylated regions (ranked by *p* value) in both RENDER-CRISPRoff treated and untreated cells and observed only modest changes in DNA methylation at unintended genomic loci compared to the target *CD55* locus (Supplementary Figs. 6, 7). Additionally, we assessed the specificity of RENDER-CRISPRoff by transcriptomic analyses using RNA sequencing (RNA-seq). Consistent with EM-seq results, the target gene *CD55* was downregulated, with the lowest *p* value in cells treated with RENDER-CRISPRoff v3 targeting *CD55*, supporting the on-target epigenome editing specificity of this approach (Fig. 3d, Supplementary Fig. 8a–c). We also performed RNA-seq on HEK293T cells treated with RENDER-CRISPRoff targeting *CLTA* and untreated controls. Consistent with previously reported DNA methylation profiles of CRISPRoff-edited HEK293T cells, the target gene *CLTA* was downregulated, with the lowest *p* value (Fig. 3e, Supplementary Fig. 8d–f). Together, our results support the high specificity of RENDER-CRISPRoff–mediated gene silencing with minimal off-target effects.

## Combinatorial epigenome editing with RENDER

A key advantage of CRISPR platforms is their ability to target multiple genes simultaneously within the same cell. We hypothesized that the RENDER-CRISPRoff platform could similarly enable combinatorial gene silencing, as demonstrated previously with CRISPRoff plasmid transfections[17]. To test this, we packaged RENDER-CRISPRoff v3 with a pool of plasmids encoding different sgRNAs targeting the promoters of *CD55* and *CD81* and delivered the particles to Jurkat and U87 MG cells (Fig. 3f). Seven days post-treatment, both cell lines showed a detectable population with both CD55 and CD81 silenced (46% for Jurkat and 82% for U87) (Figs. 3g, h). These results further demonstrate the broad use of the RENDER platform in simultaneously targeting multiple genes in a one-shot treatment.

Additionally, we aimed to enhance silencing efficacy by combining multiple sgRNAs that target the same gene promoter. To test this, RENDER-CRISPRoff v3 was packaged with either individual sgRNAs or a pool of three distinct sgRNAs targeting the same gene promoter. Fourteen days post-treatment in Jurkat cells, we observed improved silencing efficiency for *CD29*, *CD55*, and *CD81* when employing a pool of three sgRNAs in a single RENDER-CRISPRoff particle (Supplementary Fig. 9a). Together, these findings underscore the use of combinatorial RENDER to enhance target gene silencing efficiency, particularly in cases where individual sgRNAs demonstrate suboptimal activity, reinforcing its versatility for complex gene silencing applications.

## Large-scale and CRISPRoff apoprotein delivery

To evaluate whether RENDER could be applied for large-scale cell engineering, we assessed its performance across large cell populations. First, we delivered CRISPRoff to a population of $1.0 \times 10^6$ HEK293T cells at once, achieving up to over 92% silencing efficiency of CLTA-GFP, consistent with the silencing efficiency from our small-scale experiments using $1.5 \times 10^4$ cells (Fig. 3i). Furthermore, we optimized RENDER transduction for suspension cells and achieved up to 96% silencing of *CD55* and *CD81* after CRISPRoff delivery to $5.0 \times 10^5$ Jurkat cells at once, expanding from the initial $1.5 \times 10^4$ cells tested previously (Fig. 3j, Supplementary Fig. 9b, c).

CRISPR screens leverage the efficiency and flexibility of CRISPR–Cas genome editing, making them a powerful tool for biological discovery. In a typical pooled CRISPR screen, a sgRNA library is introduced into cells, with each cell receiving a different sgRNA, typically delivered via lentiviral transduction. The CRISPR–Cas protein is introduced either stably or transiently. We hypothesized that RENDER could provide an elegant approach for transiently delivering CRISPR epigenome editors in their apo-form (sgRNA-unbound proteins), which then assemble with sgRNAs inside the target cell to form functional RNP complexes, thereby expanding the methods of pooled CRISPR screens. To test this, we packaged CRISPRoff protein without sgRNA and treated CLTA-GFP HEK293T cells that constitutively express a sgRNA targeting the *CLTA* promoter (Fig. 3k). At four days post-treatment, we observe a silencing efficiency of ~93% at the highest dose, indicating successful CRISPRoff apoprotein delivery and RNP assembly within the sgRNA-expressing cells (Fig. 3l). Together, these results highlight the scalability of RENDER for applications in cell engineering and future use in high-throughput CRISPR screening where large cell quantities are required.

## Epigenome editing in primary human T cells with minimal effect on cell viability

T cells are used widely in cancer immunotherapy and frequently engineered using Cas9 nucleases to disrupt inhibitory genes. However, Cas9-mediated genome editing in T cells has been associated with adverse outcomes, including chromosomal deletions, truncations, and translocations[7]. Furthermore, delivery of genome editing reagents relies predominantly on electroporation of reagents, which requires dedicated hardware and introduces substantial cytotoxicity, elevated costs, and manufacturing burdens within cell therapy protocols[21,24,43–45].

To explore the application of RENDER-CRISPRoff v3 in primary T cell engineering, we compared its performance against CRISPRoff-encoded mRNA electroporation. Naïve T cells were first isolated from peripheral blood mononuclear cells of a healthy donor and activated for two days. The activated T cells were then either treated with a single dose of RENDER-CRISPRoff v3 targeting the *CD55* promoter or with a non-targeting control sgRNA, or electroporated with CRISPRoff mRNA and sgRNA (Fig. 4a). By two days post-treatment, we observe that RENDER-CRISPRoff treated T cells remained comparable in viability to untreated T cells, while electroporation led to approximately 40% cell death (Figs. 4b, c). Eleven days post-treatment, *CD55* is silenced in 21% of RENDER-treated T cells, while 97% of mRNA-electroporated T cells were silenced (Figs. 4d, e). Similarly, we compared silencing efficiency for *CD81* and observed that *CD81* is silenced in 35% of the T cells treated with RENDER-CRISPRoff v3 targeting *CD81*, compared to 78% for mRNA-electroporated T cells (Supplementary Fig. 10a). We further titrated the treatment dosage of RENDER-CRISPRoff v3 and observed improved silencing up to 44% of *CD55* and 60% of *CD81* (Figs. 4f, g). We only observed slight decreases (less than 5%) in cell viability at high treatment doses (Supplementary Fig. 10b, c, d). Given that electroporation leads to considerable cell loss due to its high toxicity, RENDER can achieve comparable or even more efficient epigenetic gene silencing compared to mRNA electroporation. These findings establish RENDER as a promising approach to engineer primary T cells through epigenome editing with minimal effects on cell viability as compared to mRNA electroporation.

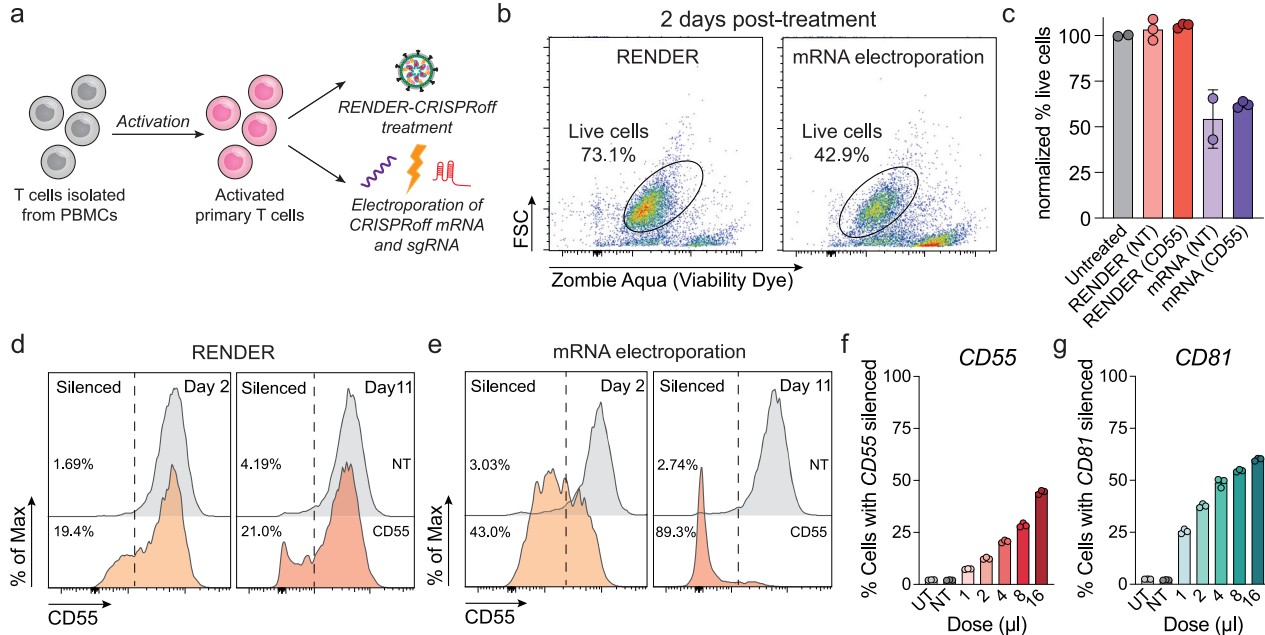

**Fig. 4 | Epigenome editing in primary human T cells. a** Schematic of RENDER-CRISPRoff treatment and electroporation of isolated primary human T cells. **b** Representative flow cytometry plot showing the viability of T cells 2 days post RENDER-CRISPRoff v3 or CRISPRoff-mRNA nucleofection with a *CD55* targeting sgRNA. **c** Quantification of live T cells 2 days post RENDER-CRISPRoff v3 or CRISPRoff-mRNA electroporation, with *CD55* targeting or non-targeting (NT) sgRNA. The data were normalized to the untreated control group. Data are shown as individual data points and mean ± SD for n = 2 (untreated cells and mRNA (NT) treated cells), or n = 3 (RENDER treated cells and mRNA (CD55) treated cells). Error bars represent the standard deviation. **d** Representative histogram plots of CD55 expression in T cells at 2 and 11 days post-treatment with RENDER-CRISPRoff v3 (5 μl) targeting *CD55* (orange) or non-targeting (gray). Percentages denote % cells with *CD55* silenced. **e** Representative histogram plots of CD55 expression in T cells at 2 and 11 days post-electroporation with CRISPRoff-mRNA and *CD55* targeting (orange) or non-targeting (gray) sgRNA. Percentages denote % cells with *CD55* silenced. **f** Quantification of *CD55* silencing in primary human T cells 5 days post-treatment with different doses of RENDER-CRISPRoff v3 with *CD55*-sgRNAs or RENDER-CRISPRoff v3 (4 μl) with non-targeting sgRNA (NT). Data are shown as individual data points and mean ± SD for n = 3 biological replicates. **g** Quantification of *CD81* silencing in primary human T cells 5 days post-treatment with different doses of RENDER-CRISPRoff v3 with *CD81*-sgRNAs or RENDER-CRISPRoff v3 (4 μl) with non-targeting sgRNA (NT). Data are shown as individual data points and mean ± SD for n = 3 biological replicates. Source data are provided as a Source Data file.

## Therapeutic epigenome editing in iPSC-derived neurons

Many neurodegenerative diseases are caused by the toxic accumulation of mutant proteins, positioning them as attractive candidates for therapeutic genome editing aimed at perturbing the underlying mutated genes. However, conventional genome editing in neurons is challenging due to unique mechanisms in DNA double-strand break (DSB) repair[46]. A single treatment that could robustly and durably silence disease-associated genes in neurons without inducing DSBs would offer a powerful alternative for therapeutic intervention.

To generate programmed epigenetic repression of endogenous genes in neurons, we previously expressed CRISPRoff transiently in induced pluripotent stem cells (iPSCs) via plasmid DNA transfection, followed by neuronal differentiation. The high efficacy of RENDER across diverse cell types motivated us to test the delivery of epigenome editor RNPs directly into neurons to repress endogenous genes, thereby bypassing the requirement of pre-editing in iPSCs. We first packaged CRISPRoff with sgRNAs targeting the *CD81* promoter and treated neurons 10 days post-differentiation from WTC-11 iPSCs (Fig. 5a). Seven days post treatment, we observe robust silencing of *CD81* in treated neurons, with a 79% reduction in total CD81 protein levels compared to the non-targeting control (Figs. 5b, c). To test the durability of epigenetic repression in neurons, we also measured CD81 levels two weeks after CRISPRoff treatment and observed that CD81 remains stably repressed in the majority of treated neurons (Figs. 5b, c). These results demonstrate that RENDER mediates robust target gene repression in iPSC-derived neurons, thereby enabling us to directly engineer customized neurons with target genes repressed durably.

Silencing of the *MAPT* gene, which encodes Tau protein, is of great therapeutic interest for treating neurodegenerative diseases. Studies have shown that reducing *MAPT* expression protects against cognitive deficits in Alzheimer's disease model mice[47]. Additionally, *MAPT* missense mutations directly contribute to tauopathies, such as the V337M mutation that is associated with frontotemporal dementia (FTD). Current therapeutic strategies, including small interfering RNA (siRNA), antisense oligonucleotides (ASOs), and antibodies, require regular re-dosing to maintain efficacy. In contrast, the RENDER platform offers a promising approach to achieve durable repression of Tau expression with a single treatment. To investigate the potential therapeutic use of RENDER in neuronal contexts, we delivered CRISPRoff with sgRNAs targeting the *MAPT* promoter in iPSC-derived neurons harboring the *MAPT* V337M mutation (Fig. 5a). Eight days post-treatment with a high dose of CRISPRoff, we observed an approximate 63% reduction in Tau protein expression in the treated neurons, compared to non-targeting control (Figs. 5d, e). Subsequent analysis at 15 days post-treatment showed a 51% reduction in Tau protein levels in neurons treated with a high dose of CRISPRoff, indicating long-lasting repression of Tau protein expression after a single treatment. Collectively, these results underscore the ability of our RENDER platform to effectively deliver CRISPR-based epigenome editors into challenging cell types, such as neurons, for targeted epigenetic modulation in translational research and therapeutic applications.

## Discussion

Here, we present RENDER as a robust platform for transient delivery of CRISPR-based epigenome editor ribonucleoproteins into human cells

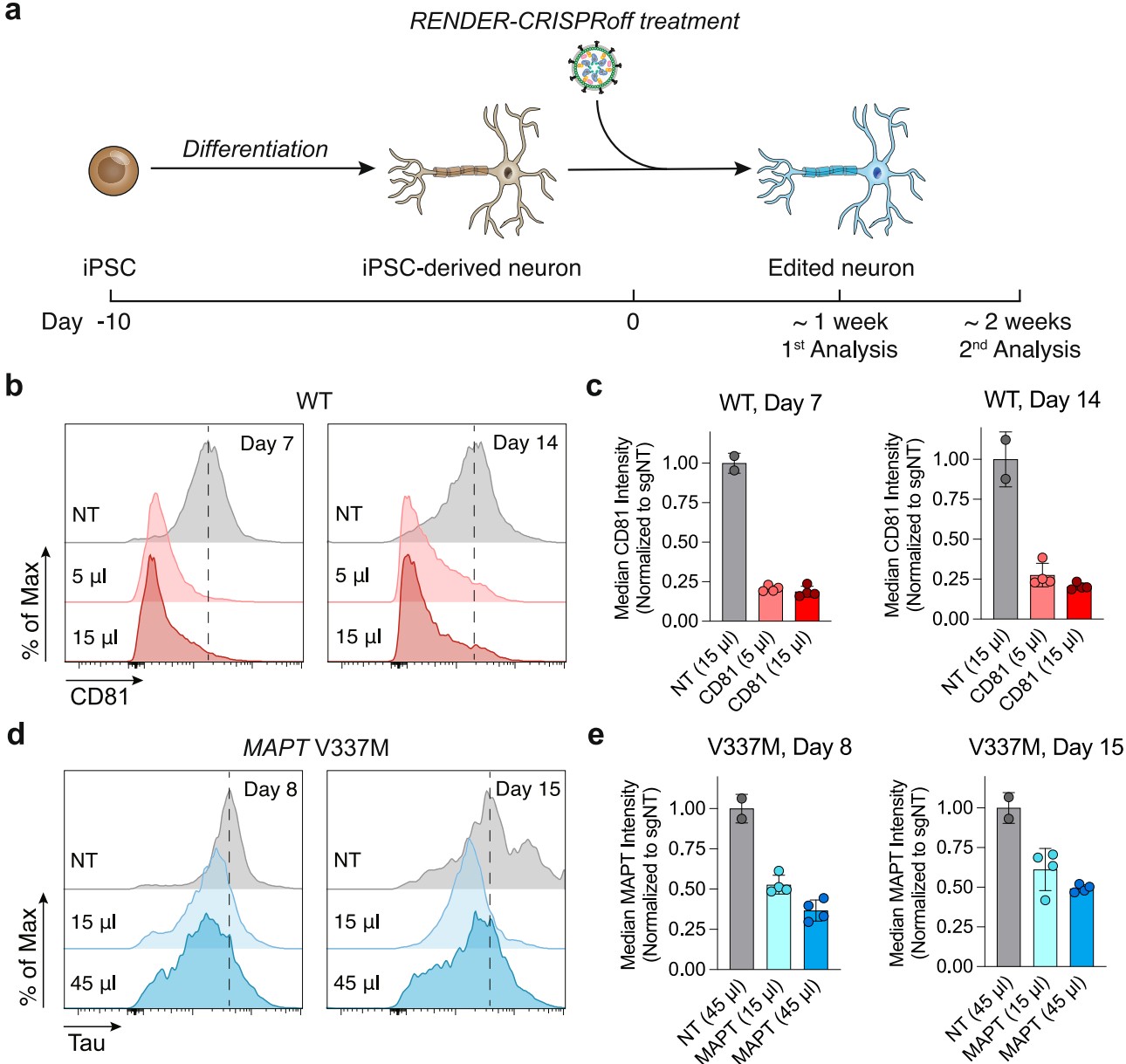

**Fig. 5 | Epigenome editing iPSC-derived neurons with RENDER. a** Experimental workflow of gene silencing by RENDER-CRISPRoff v3 in iPSC-derived neurons. **b** Representative histogram plots of CD81 expression in iPSC-derived neurons (wild-type, WT) at day 7 and 14 post-treatment with different doses of RENDER-CRISPRoff v3 targeting *CD81* or non-targeting (NT). **c** Quantification of CD81 expression in iPSC-derived neurons (wild-type, WT) at day 7 and 14 post-treatment with different doses of RENDER-CRISPRoff v3 targeting *CD81*. CD81 expression is measured by antibody staining and flow cytometry and normalized to non-targeting (NT) control. Data are shown as individual data points and mean ± SD for n = 2 (RENDER-CRISPRoff v3 NT treated neurons), or n = 4 (RENDER-CRISPRoff v3 targeting *CD81* treated neurons). Error bars represent the standard deviation.

**d** Representative histogram plots of Tau expression in iPSC-derived neurons (V337M mutant) at day 8 and 15 post-treatment with different doses of RENDER-CRISPRoff v3 targeting *MAPT* or non-targeting (NT). **e** Quantification of Tau expression in iPSC-derived neurons (V337M mutant) at day 8 and 15 post-treatment with different doses of RENDER-CRISPRoff v3 targeting *MAPT*. Tau expression is measured by antibody staining and flow cytometry and normalized to non-targeting (NT) control. Data are shown as individual data points and mean ± SD for n = 2 (RENDER-CRISPRoff v3 NT treated neurons), or n = 4 (RENDER-CRISPRoff v3 targeting *MAPT* treated neurons). Error bars represent the standard deviation. Source data are provided as a Source Data file.

(Supplementary Fig. 11). RENDER can package diverse epigenome editors as cargo, including CRISPRi for short-term, reversible gene repression and CRISPRoff and DNMT3A-3L-dCas9 for durable gene silencing. RENDER takes advantage of the eVLP platform that has been established recently to deliver CRISPR nucleases and base editors[30]. We further optimized the RENDER platform specifically for CRISPRoff to increase protein and sgRNA cargo packaging, which improved its epigenetic silencing potency by 4.4-fold, yielding RENDER-CRISPRoff v3. A single dose of RENDER-CRISPRoff v3 induces efficient and durable epigenetic silencing of target genes for over a month and is functional across multiple cell types and different gene targets.

Through our experiments, we demonstrate the robust applicability and potency of RENDER across various cell types, addressing prior challenges in delivery while further improving editing efficiency. While CRISPRoff and other epigenome editors have been delivered as DNA in AAV vectors or mRNA in LNPs recently[18,19,39], our work reports the direct RNP delivery of epigenome editors. RNP delivery offers several advantages, including its transient nature and minimized off-

target editing[30,32,48–50]. Our results demonstrate that RENDER effectively addresses these challenges, providing a more efficient solution for epigenome editor delivery. Plasmid DNA transfections or lentiviral infection of CRISPRi/off typically yield suboptimal transfection efficiencies, often necessitating additional selection methods (e.g., cell sorting or antibiotic selection) to isolate a pure population of editor-carrying cells. Additionally, mRNA electroporation is highly dependent on reagents and equipment, and inevitably reduces cell viability. Furthermore, these methods are often not suitable for key cell types, such as neurons.

Our data show that RENDER-based delivery of CRISPRoff results in robust silencing of endogenous genes in cultured human neurons, including notable repression of Tau protein expression in neurons harboring a known pathogenic mutation. These results highlight the therapeutic potential of RENDER-CRISPRoff as a strategy for treating neurodegenerative diseases. Recent studies have shown the promise of virus-like particles (VLPs) as vehicles for in vivo gene editor RNP delivery[27,30–32,48–51], and the RENDER platform extends this approach to epigenome editors, laying the groundwork for future in vivo epigenetic therapies. Furthermore, recent advancements in envelope protein engineering have further shown the feasibility of reprogramming virus/virus-like particle targeting tropism to specific cell types in complex environments, including in vivo, supporting the potential of RENDER for tissue-specific delivery and epigenome editing[23,27]. Beyond advances in epigenome editor RNP delivery, RENDER's high efficacy in CRISPRoff apoprotein delivery can offer a potential platform for CRISPR screens in sgRNA library-expressing cell lines, facilitating functional genomics applications through epigenome editing perturbations. Thus, RENDER expands the potential of programmable epigenome editors for both biomedical research and therapeutic applications.

# Methods

## Molecular cloning
All plasmids to express protein components were cloned using Gibson assembly. DNA was PCR amplified with KAPA HiFi HotStart ReadyMix PCR Kit (Roche). Plasmids were transformed into Top10 chemically competent *Escherichia coli* (Thermo Fisher Scientific) and were prepared using either QIAprep Spin Miniprep kits (Qiagen) or ZymoPURE II Plasmid Maxiprep Kit (Zymo Research). CRISPRoff (KOX1), CRISPRi (KOX1), DNMT3A-3L-dCas9, TET1-dCas9 and dCas9 sequences were PCR amplified from previous constructs[17] (Addgene #167981 and #167983) and cloned in to the eVLP backbone from pCMV-MMLVgag-3×NES-ABE8e (Addgene #181751) to replace the base editor ABE8e (Supplementary Table 1). The ZIM3 KRAB sequence was obtained from previous study[36], synthesized as a gBlock (IDT) and cloned into the CRISPRoff and CRISPRi construct to replace the KOX1 KRAB domain. Similar to CRISPRoff, all editors include a direct fusion of HA-tag, 2×SV40 NLS and BFP to the C-terminus of dCas9.

The transient sgRNA expression plasmid encodes for expression of a sgRNA from a modified U6 (mU6) promoter along with expression from an EF1α promoter of a puromycin resistance gene and mCherry separated by a T2A cleavage sequence. To generate a minimal plasmid backbone, the origin of replication and the ampicillin resistance gene from pUC19 and the terminator from the pCAG mammalian expression vector were PCR amplified with KAPA HiFi HotStart ReadyMix PCR Kit (Roche). The lentiviral pLG1 library vector (Addgene #217306) was digested with XbaI and EcoRI to obtain only the mU6-sgRNA and EF1α-puroR-T2A-mCherry sequence. The transient plasmid was constructed using Gibson assembly of these three fragments with NEBuilder® HiFi DNA Assembly (NEB).

To change the protospacers, the transient backbone was digested with BstXI and BlpI, which cut downstream of the mU6 promoter. The protospacer sequence (Supplementary Table 2) was then ordered as two complementary oligos (IDT) with compatible BstXI and BlpI

overhangs and ligated into the cut vector. The protospacer sequences were chosen based on previous algorithms to predict active CRISPRi sgRNAs targeting gene promoters[1]. Additionally, a protospacer sequence targeting the *GAL4* gene in *Saccharomyces cerevisiae* was used as non-targeting sgRNA (Supplementary Table 2).

## Tissue culture
All cell lines were obtained and authenticated by the UC Berkeley Cell Culture Facility. The CLTA-GFP and HIST2H2BE-GFP HEK293T cell lines originated from previous study[38]. Lenti-X, HEK293T, HeLa and HepG2 cells were cultured in DMEM (Gibco). Jurkat and K562 cells were cultured in RPMI-1640 (Gibco). h-TERT RPE-1 cells were cultured in DMEM/F-12, GlutaMAX™ (Gibco). U87 and SH-SY5Y cells were cultured in DMEM, supplemented with 1% (v/v) MEM Non-Essential Amino Acids Solution (Gibco) and 1 mM sodium pyruvate (Gibco). All media was supplemented with 10% (v/v) FBS (VWR) and 100 U/ml streptomycin, 100 mg/ml penicillin (Gibco). Cell lines were cultured at 37 °C with 5% $CO_2$ in tissue culture incubators and routinely confirmed to be negative for mycoplasma.

## Generation of stable sgRNA expressing cell line
To test the sgRNA-free CRISPRoff-eVLP, CLTA-GFP HEK293T reporter cells were transduced with lentivirus to constitutively express a sgRNA targeting the *CLTA* promoter. Lentiviral particles were produced by transient transfection of HEK293T cells. Cells were seeded at a density of $4.0 \times 10^5$ cells per well in a 6-well BioLite Microwell plate (Thermo Fisher Scientific) and co-transfected the next day with standard packaging plasmids (0.1 μg gag-pol, REV, TAT, 0.2 μg VSVG) and 1.5 μg lentiviral sgRNA expression vector (Addgene #217306) using Opti-MEM (Gibco) and TransIT®-LT1 Transfection Reagent (Mirus Bio), following the manufacturer's protocol. Media was changed 24 h post-transfection with complete DMEM. Viral supernatants were harvested 48–60 h after transfection and filtered through a 0.45 μm PES syringe filter (VWR). CLTA-GFP HEK293T cells were transduced with lentiviral particles and subsequently sorted on mCherry expression to obtain a > 99% sgRNA expressing cell line.

## Epigenome editor eVLP production and concentration
Epigenome editor eVLPs were produced by transient transfection of Lenti-X 293T cells. Cells were seeded at a density of $4 \times 10^6$ cells per 10-cm BioLite Cell Culture dish (Thermo Fisher Scientific) and transfected the next day using Opti-MEM (Gibco) and TransIT®-LT1 Transfection Reagent (Mirus Bio), following the manufacturer's protocol.

For the comparison of different epigenome editor eVLPs, cells were co-transfected with 0.5 μg VSV-G (Addgene #8454), 3.0 μg gag-pol (Addgene #35614), 1.5 μg gag-editor and 5.0 μg mU6-sgRNA plasmids. For stoichiometry optimization, the total amount of gag-CRISPRoff (ZIM3) and gag-pol plasmid was fixed at 4.5 μg, while the ratio of gag-CRISPRoff (ZIM3) was varied from 10 to 100%. For sgRNA optimization, a fixed stoichiometry of 90% gag-CRISPRoff (ZIM3) (4.05 μg) and 10% gag-pol (0.45 μg) was used, while the amount of mU6-sgRNA plasmid was varied from 1.0 to 6.0 μg.

For packaging of the final optimized CRISPRoff-eVLP v3, cells were co-transfected with 0.5 μg VSV-G, 0.45 μg gag-pol, 4.05 μg gag-CRISPRoff (ZIM3) and 6.0 μg mU6-sgRNA. For multiplexed gene targeting, plasmids expressing different sgRNAs were pooled for co-transfection, keeping the total amount of sgRNA plasmid at 6.0 μg. For protein-only CRISPRoff-eVLP v3, cells were co-transfected with 1.0 μg VSV-G, 0.90 μg gag-pol and 8.1 μg gag-CRISPRoff (ZIM3).

Three days post transfection, editor-eVLP containing supernatant was collected, filtered through a 0.45 μm PES syringe filter (VWR) and precipitated overnight using Lenti-X Concentrator (Takara Bio) according to manufacturer's instructions. Editor-eVLPs were concentrated 100-fold by resuspending in 100 μl Opti-MEM, frozen at a

rate of −1 °C/min, and stored at −80 °C. Frozen eVLPs were thawed on ice immediately before use.

## CRISPRoff-eVLP Cas9 protein quantification

CRISPRoff-eVLPs were lysed in Laemmli sample buffer (50 mM Tris-HCl pH 7.0, 2% sodium dodecyl sulfate (SDS), 10% (v/v) glycerol, 2 mM dithiothreitol (DTT)) by heating at 95 °C for 15 min. The concentration of CRISPRoff protein in concentrated CRISPRoff-eVLPs was quantified using the FastScan™ Cas9 (*S. pyogenes*) ELISA kit (Cell Signaling Technology; 29666 C) according to the manufacturer's protocols. Recombinant Cas9 (*S. pyogenes*) nuclease protein (New England Biolabs; M0386S) was used to generate the standard curve for quantification.

## Editor-eVLP transduction and flow cytometry

For adherent cells, $1.5 \times 10^4$ cells were plated for transduction in 200 µl media per well in a 96-well plate (Corning). After 18–24 h, editor-eVLPs were directly added to the media. Suspension cells were plated at $1.5 \times 10^4$ cells in 200 µl media per well and directly transduced with editor-eVLPs by centrifugation at $1500 \times g$ for 1.5 h at 32 °C.

For scaling-up eVLP transduction in Jurkat cells, non-treated 6-well or 96-well plates (Falcon) were coated with RetroNectin (Takara Bio) at a density of 8 µg/cm², following manufacturer's instructions. Jurkat cells were plated at $1.5 \times 10^4$ cells in 100 µl media per 96-well or $5.0 \times 10^5$ cells in 1.5 ml per 6-well and transduced with eVLPs by centrifugation at $1500 \times g$ for 1.5 h at 32 °C. 18–24 h post-transduction, 100 µl fresh media was added per 96-well or 1.5 ml per 6-well.

Treated cells were passaged every 2–3 days and durable target gene silencing was assessed by flow cytometric analysis. Protein expression was assessed by cell surface antibody staining of live cells. Cells were incubated with diluted antibody (Supplementary Table 1) for 30–60 min in the dark at room temperature, washed once with PBS and measured on the BD FACSymphony A1 Cell Analyzer (BD Biosciences). For the different GFP-tagged HEK293T reporter cell lines, protein expression was directly assessed by flow cytometry. Data analysis was performed using FlowJo (v10.10).

## Isolation, activation, eVLP transduction and mRNA electroporation of primary human T cells

Cryopreserved human peripheral blood mononuclear cells (STEMCELL Technologies) were thawed and T cells were isolated using EasySep Buffer and the EasySep Human T Cell Isolation Kit (STEMCELL Technologies) according to manufacturer's instructions. Naïve T cells were cultured in basal media comprised of X-VIVO 15 Serum-free Hematopoietic Cell Medium (Lonza) with 5% FBS (v/v) (VWR), 10 mM *N*-Acetyl-L-cysteine (Sigma-Aldrich) and 50 µM 2-mercaptoethanol (Gibco), or frozen for long-term storage in basal media, without FBS, supplemented with 10% DMSO.

Isolated or thawed naïve T cells were kept in basal media for 24 h and then activated by adding Dynabeads Human T-Activator CD3/CD28 (1:1 bead-to-cell ratio, Gibco), 300 U/ml recombinant human IL-2 (Peprotech), 5 ng/ml recombinant human IL-7 (Peprotech) and 5 ng/ml recombinant human IL-15 (R&D Systems). After two days of activation, magnetic beads were removed and T cells were cultured in basal media with 500 U/ml IL-2, 5 ng/ml IL-7 and 5 ng/ml IL-15.

For eVLP transduction, T cells were seeded at $5.0 \times 10^4$ cells in 100 µl media per well and 5 µl of CRISPRoff-eVLPs packaged with a pool of three sgRNAs for CD55 (a, b, c), or two sgRNA for CD81 (a, b) or a non-targeting (NT) sgRNA (Supplementary Table 2) were directly added to the media.

For mRNA electroporation, $1.0 \times 10^6$ T cells were nucleofected with 1 µg of CRISPRoff mRNA (Aldeveron) and 2.5 µg of synthetic sgRNA in P3 Primary Cell 96-well Nucleofector kit as previously described[52]. Modified synthetic sgRNAs were ordered from Synthego with the same protospacer sequences for CD55 (a, b) or CD81 (a, b) and

equally pooled for electroporation, or a non-targeting protospacer sequence (NT) (Supplementary Table 2).

Treated T cells were cultured in basal media with 500 U/ml IL-2, 5 ng/ml IL-7 and 5 ng/ml IL-15 and passaged every 2–3 days. Target gene silencing was assessed by flow cytometric analysis, as described above, in combination with a viability dye (Supplementary Table 3).

## Differentiation and transduction of iPSC-derived neurons

Neurons were derived from WTC-NGN2 iPSCs following a differentiation protocol as previously described[46]. On the third day of differentiation (Day −7 relative to RENDER treatment), neurons were seeded onto PDL-coated culture plates (Corning #356414). In 24-well plate format, each well contained ~100,000 cells and was treated with 5, 15, or 45 µl of eVLP-containing media. In larger plate formats, these ratios were scaled up proportionally to the feeding volume.

Neurons were fixed with a Zinc fixation buffer (0.1 M Tris HCl pH 6.5, 0.5% ZnCl₂, 0.5% Zn Acetate and 0.05% CaCl₂) at 4 °C overnight. For CD81 staining, the second day, after removing the fixation solution, neurons were washed twice with PBS, then incubated with APC anti-human CD81 (TAPA-1) antibody solution (1.25:100) for 1 h at RT. For Tau protein staining, the second day, after removing the fixation solution, neurons were washed twice with PBS, then incubated and permeabilized with permeabilization/blocking buffer (10% Goat Serum, 1% TBS, 3% BSA, 1% glycine and 0.5% Tween-20) at RT for 30 min. After removing this solution and washing 2 times with PBS, neurons were incubated with the following buffers at RT, with two PBS washes after each incubation: primary antibody solution (1:1000, #sc-21796) for 1 h, secondary antibody solution (1:500, #ab150117) for 1 h. After removing staining solution and washing 2 times with PBS, neurons were then pelleted, resuspended in 300 µl of PBS per sample, and triturated gently to singularize. These samples were passed through strainer-capped FACS tubes (eg Stellar Sci #FSC-9005), and analyzed on an Attune NxT (Thermo Fisher Scientific). Results were interpreted using FlowJo (v10.10).

## Western blot analysis

Concentrated RENDER-CRISPRoff (v3) were denatured at 90 °C for 5 min in reducing Laemmli SDS sample buffer (Thermo Fisher Scientific). Denatured proteins were separated on a NuPAGE 3–8% Tris-Acetate Gel (Thermo Fisher Scientific) in NuPAGE Tris-Acetate SDS Running Buffer (Thermo Fisher Scientific) at 150 V for 1 h. Proteins were wet transferred into PVDF membrane (Bio-Rad) in Tris-Glycine buffer (25 mM Tris-HCl, 250 mM glycine, 20% (v/v) ethanol) at 300 mA for 2 h. The membrane was washed three times with TBS-T (20 mM Tris-HCl, 150 mM NaCl, 0.1% Tween-20), blocked with 5% nonfat dry milk (Apex) in TBS-T at room temperature for 1 h and incubated at 4 °C overnight with mouse-anti FLAG (1:400, Sigma-Aldrich, F3165) or mouse-anti Cas9 (1:1,000, Active Motif, AB_2793684) in 1% milk TBS-T with gentle rocking. After three washes with TBS-T, the membrane was incubated at room temperature for 1 h with goat anti-mouse secondary antibody (1:10,000, LI-COR, IRDye 680RD) in TBS-T by rocking. The membrane was washed three times with TBS-T and imaged on the Odyssey CLx (LI-COR).

## Whole genome DNA methylation sequencing and analysis

We generated whole genome enzymatic methyl sequencing (WGEMS) libraries for 4 samples corresponding to two technical replicates of wild-type (untreated) Jurkat cells and Jurkat cells treated with RENDER-CRISPRoff targeting *CD55*. Genomic DNA was isolated using the PureLink™ Genomic DNA Mini Kit (Invitrogen), 200 ng of DNA was diluted with TE buffer to 108 µl. Control DNA Unmethylated Lambda (NEB #E7122AVIAL) and Control DNA CpG methylated pUC19 (NEB #E7123AVIAL) were diluted 1:50 with TE buffer and 1 µl was added to the genomic DNA (final volume 110 µl). DNA samples were sheared using a Covaris S220 (peak incident power: 175 W, duty factor: 10%,

cycles per burst: 200, treatment time: 180 s). 50 μl of sheared DNA was used for enzymatic methyl conversion following the NEBNext® Enzymatic Methyl-seq Kit (NEB #E7120S/L) for standard libraries using the formamide denaturation protocol. 5 ng of deaminated DNA was PCR-amplified for 6 cycles, followed by PCR-clean up and eluted in TE buffer according to protocol. Amplified libraries were quantified on an Agilent 2100 TapeStation system (Agilent) and equally pooled to a final concentration of 2.475 ng/μl in 50 μl (approximately 10 nM at 375 bp). The prepared libraries were sequenced paired-end 150 bp reads on an Illumina NovaSeq X 1.5B 2 Lanes (1.5 billion clusters). We obtained between 338.2 M to 439.6 M paired-end reads per library across all 4 libraries.

Reads were trimmed using cutadapt (version 4.9)[53], filtering empty resulting reads (-m 1) and specifying both forward (-a AGATCGGAA-GAGCACACGTCTGAACTCCAGTCA) and reverse (-A AGATCGGAA-GAGCGTCGTGTAGGGAAAGAGTGT) adapters. The resulting reads were aligned using Bismark (version 0.24.2)[54] to the hg38 reference genome with methylated and unmethylated control DNA sequences provided by NEBNext (https://neb-em-seq-sra.s3.amazonaws.com/grch38_core%2Bbs_controls.fa). The Bismark tools bismark_methylation_extractor and bismark2bedGraph were then used to create a methylation report, the top and bottom strand reads were combined (on the basis of CpG symmetry), and loci that had less then 10 mapped reads across the 8 methylation reports (top and bottom strands, four samples) were filtered out (approximately 2% of loci). Differential methylation analysis was performed using bsseq (version 1.40.0)[55] and DSS (version 2.52.0)[56] in R (version 4.4.1) by calling DMLtest with 500 bp smoothing. Plots were generated using Python (version 3.12.6) with Matplotlib (version 3.9.2)[57], NumPy (version 2.1.1)[58], and Polars (version 1.9.0).

### RNA sequencing and data analysis

CLTA-GFP HEK293T cells were maintained for 24 days post-treatment of RENDER-CRISPRoff v3 targeting *CLTA*. Jurkat cells were maintained for 24 days post-treatment of RENDER-CRISPRoff v3 targeting *CD55*. Cells were centrifuged at 500 × g for 5 min and washed with PBS. Total RNA was extracted using RNeasy Mini Kit (QIAGEN). Poly(A) mRNA was enriched using poly(A) mRNA Magnetic Isolation Module (New England Biolabs). Library preparations were carried out using NEBNext Ultra II Directional RNA Library Prep Kit for Illumina (New England Biolabs), starting with 1000 ng total RNA. Final libraries were assessed using an Agilent 2100 TapeStation system (Agilent), and sequenced as paired end 150 base pair reads on a NovaSeq X (Illumina). Reads were trimmed using cutadapt (version 4.9)[53], filtering empty resulting reads (-m 1), poly(A) tails (--poly-a), and specifying both forward (-a AGATCGGAA-GAGCACACGTCTGAACTCCAGTCA) and reverse (-A AGATCGGAA-GAGCGTCGTGTAGGGAAAGAGTGT) adapters. The resulting reads were quantified using kallisto (version 0.51.1)[59], with an index built from the Ensembl v114 cDNA sequences for transcripts for the human genome (GRCh38.p14). Transcript-level counts were aggregated to gene-level counts using tximport (version 1.34.0) with a transcript-to-gene table built from Ensembl v114 using BioMart (version 2.62.1)[60] filtered to only protein-coding genes, both in R (version 4.4.2). Differential expression analysis was then performed using DESeq2 (version 1.46.0)[61] with independent filtering off (also in R, version 4.4.2). Finally, plots were generated using Python (version 3.13.2) with Matplotlib (version 3.10.1)[57], NumPy (version 2.2.5)[58], and SciPy (version 1.15.3)[62].

### Statistics & reproducibility

Statistical parameters including the definitions and exact value of n (e.g., total number of experiments and replications), deviations, *p* values, and the types of the statistical tests are reported in the figures and corresponding figure legends. No statistical methods were used to predetermine sample size. RNA-seq and methylation sequencing data were analyzed as described above. All other statistical analyses were conducted using Prism 10 for macOS (GraphPad), Excel for MacOS (Microsoft). Unless otherwise specified in the figure legends, all data analyses were performed using unpaired t-tests for pairwise comparisons.

### Reporting summary

Further information on research design is available in the Nature Portfolio Reporting Summary linked to this article.

## Data availability

Plasmids generated for this study have been deposited in the Addgene database [https://www.addgene.org/James_Nunez/]. The bioinformatic datasets for the WGEM-seq and RNA-seq experiments are available in the Sequence Read Archive (SRA) of the National Center for Biotechnology Information (NCBI) under the accession number PRJNA1195923. Source data are provided with this paper.

## Code availability

Custom code developed for this study, along with a valid license, is available via GitHub at https://github.com/Nunez-Lab.

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

## Acknowledgements

We thank Kevin Wasko for technical assistance and members of the Nuñez lab for helpful discussions. This work was funded by research grants awarded to J.K.N.: NIH R35 MIRA (R35GM155044-01), CRISPR Cures for Cancer Initiative (Gladstone Institutes, UCB 20240097), Bakar

Labs (51474), and an Innovation Award from the Laboratory for Genomics Research (UCB 059006-001). J.K.N. and S.E.C. are Investigators of the Chan Zuckerberg Biohub San Francisco. S.B. is funded by internship grants from the Nijbakker-Morra Stichting and Peter Grootenhuis Life Sciences. G.N.R. is funded by the Gladstone-CIRM Postdoctoral Fellowship. Neuron work was enabled by the Gladstone Stem Cell and Flow Cytometry Cores. J.P.L, J.I.B. and I.J.O. are funded by Graduate Research Fellowships from the National Science Foundation. R.K.P. is funded by graduate fellowships from the University of California Cancer Research Coordinating Committee and the Shurl and Kay Curci Foundation. I.J.O. is funded by a University of California, Berkeley Mentored Research Award and a Gilliam Fellowship from the Howard Hughes Medical Institute. J.I.B. is funded by the National Institutes of Health Genetic Dissection of Cells and Organisms Training Program (T32GM132022). BRC was supported by the National Institutes of Health (R01-AG072052, R01-HL130533, R01-HL13535801, P01-HL146366), the California Institute for Regenerative Medicine (INFR6.2-15527), and The Charcot-Marie-Tooth Association. BRC acknowledges support through a gift from the Roddenberry Foundation and Pauline and Thomas Tusher.

## Author contributions

D.X., S.B. and J.K.N. led the study, designed the experiments and wrote the manuscript with assistance from the co-authors. D.X. and S.B. performed the majority of the RENDER experiments. G.N.R. and P.H.D. designed and performed the neuron editing experiments with supervision from B.R.C. J.P.L. and P.J.C. performed the WGEM-seq experiments and J.P.L. designed and analyzed the computational analysis pipeline under the supervision of S.E.C. R.K.P. and C.D.N. performed the T cell experiments. J.I.B. performed the TET1-dCas9 reactivation experiments and assisted with testing RENDER in different cell types. A.E.C. constructed the sgRNA expression plasmid and performed the H2B silencing experiments. I.J.O. assisted with computational data analysis.

## Competing interests

J.K.N. and D.X. have submitted a patent application to U.S. Provisional Patent Office related to the delivery of CRISPR components, as part of this work (application No. 63/686,584). J.K.N. is an inventor of patents related to the CRISPRoff/on technologies, filed by The Regents of the University of California. B.R.C. is a founder of Tenaya Therapeutics and holds equity in the company. The remaining authors declare no competing interests.
