## [Transparent Peer Review file · Nature Communications]

Programmable epigenome editing by transient delivery of CRISPR epigenome editor ribonucleoproteins

Corresponding Author: Dr James Nuñez

Version 0:

Reviewer comments:

Reviewer #1

(Remarks to the Author)

In the present manuscript, the authors described the development of Robust Enveloped Delivery of Epigenome-editor Ribonucleoproteins (RENDER), that delivers epigenetic editors as RNP by packaging and delivering epigenetic editor cargoes in the VLPs. The authors optimized the RENDER for improved cargo packaging, and demonstrated RENDER-mediated epigenetic editing in primary T cells and iPSC-derived neurons. The effort to deliver epigenetic editors transiently as RNP cargo is highly commendable. Additional demonstration of the reversibility of epigenome editing, and multiplexability of RENDER platform also adds significant value to the study. The manuscript is generally well-written, with concise and accurate language. The figures and data support the conclusions effectively, and the experiments conducted are comprehensive and well-reasoned. I recommend the manuscript for publication if the authors can sufficiently address the following comments.

Comments:

- Primary T cell targeting – Have the authors tested multiple doses for T-cell transduction? The Methods section indicates that only 5 μ L of 100 \times concentrated RENDER was used for T-cell transduction, which appears to be a very low dose for a hard-to-transduce cell type. Was this dose chosen due to concerns about toxicity? If possible, it would be valuable for the authors to test multiple doses and provide a dose-response curve. This would also allow for a comparison of editing efficiency with mRNA electroporation, enhancing the comprehensiveness of the study.
- eVLP production method – The manuscript describes collecting VLP supernatant three days post-transfection, while many existing studies collect VLPs 40–48 hours post-transfection (e.g., Banskota and Raguram et al., 2022; An et al., 2024; Hamilton et al., 2024; Mangeot et al., 2019). Have the authors performed a side-by-side comparison of these collection times? If not, could the authors clarify the rationale for this deviation from standard protocols?
- Line 163-165 – The authors conclude that lower doses of RENDER-mediated delivery of CRISPRoff result in reduced durability. This observation is rather intriguing, as durability would typically depend on the editor itself rather than the delivery method or dose. Can the authors speculate on the underlying mechanism for this phenomenon?
- The study demonstrates efficient editing with RENDER in cell culture. To increase the significance of the manuscript, it is highly recommended that the authors explore the in vivo translatability of the platform. Demonstrating RENDER-mediated delivery of epigenetic editors in mouse models would add substantial impact to the study.

Reviewer #2

(Remarks to the Author)

In their article, “Programmable Epigenome Editing by Transient Delivery of CRISPR Epigenome Editor Ribonucleoproteins”, Da Xu et al. propose and extensively demonstrate how their RENDER platform efficiently delivers CRISPR epigenome editors to various human cells, including primary T cells and neuronal cells, which are typically difficult to transfect. The study is experimentally robust and of considerable scientific interest. However, there are several points that require further clarification.

Figure 1: The authors primarily focus on inducing gene downregulation using an endogenously GFP-tagged gene as their target. Only a single experiment was conducted using an editor enzyme responsible for increasing gene expression, TET (Fig. 1i-j). The efficiency of gene activation demonstrated is significantly lower than that of downregulation. The authors

state: "Further optimization experiments are needed to address this issue" (lines 154-157). However, the authors should provide a more detailed explanation for the low activity of this construct as in their previous study (doi: 10.1016/j.cell.2021.03.025), plasmid transfection achieved stable reactivation in 74% of the cells. The authors should elaborate on the advantages of using the RENDER platform despite the reduced efficiency compared to plasmid transfection. Additionally, what percentage of activation would be achieved if another endogenous gene was targeted?

Figure 3a: The authors report low silencing efficiency in K562 cells, attributing this to cell type limitations (lines 206-207). Did they attempt to assess differential uptake or to optimize delivery conditions, for example, by increasing the doses of the editor? Such experiments would strengthen their conclusions.

Figure 3b: The authors present genome-wide methylation data to demonstrate the high specificity of the approach. However, they state (lines 218-221) that "modest" methylation events unrelated to the CD55 target were detected. It would be important to determine whether these slight off-target methylations are associated with biological effects. The authors should include gene expression data to assess the potential functional impact of these off-target effects.

Figure 4: The authors compare the efficiency of editing via the RENDER platform to that of mRNA nucleofection. They highlight the RENDER platform's advantage in maintaining greater cell viability compared to mRNA transfection via electroporation (lines 67-69, 347-349). While higher cell viability is a benefit, the data in Fig. 4d-e show that mRNA transfection results in significantly higher silencing of CD55 cells at both 2 days (43.7%) and 11 days (97%), compared to 24% and 21% with RENDER. The authors should emphasize why the RENDER platform should be preferred over mRNA, perhaps by discussing other advantages such as cost or preparation time.

Furthermore, if the silencing efficiency of RENDER for endogenous genes such as CD55 and CD81 is between 20% and 35%, the authors should clarify its advantage over plasmid DNA. They note (lines 644-646) that plasmid DNA requires cell selection methods, such as sorting or enzymatic selection, due to low transfection and silencing efficiency. However, as seen in Fig. 4d and 4f, RENDER's silencing efficiency also appears low and may similarly require additional cell selection steps, depending on the gene. This point should be addressed in the discussion.

Extended Data Figures 4 and 5: The statement that methylation was "gained" in untreated cells appears unclear. The authors are advised to revise this sentence and provide a more detailed explanation of the figures.

Minor comments:

Paragraph "Multiplexed Epigenome Editing..." (line 244): The use of terms such as "multiplexed" and "multiple" to describe targeting of only 2 genes seems overstated. The authors might rephrase this section for accuracy.

Line 255-258: The concept of the apo-form should be explained more clearly.

Materials and Methods: The article does not provide details on how mRNA was obtained or the conditions used for the electroporation reaction. These details should be included.

Terminology Consistency: Throughout the article, the term "RENDER-CRISPRoff v3" should be used consistently wherever these conditions are applied (e.g., line 200).

Reviewer #3

(Remarks to the Author)

Reviewer #4

(Remarks to the Author)

The manuscript by Xu D. et al. developed viral-like particles (VLPs) to deliver epigenome-editor RNPs named RENDER. The main focus was on quantitative assessments of gene silencing in several types of human cell lines using the RENDER-CRISPRoff platform. This platform was used to knock out CD55/CD81 in primary T cells and also to silence CD81 and tau expression through targeted MAPT mutation at V337M in iPSC-derived neurons. However, more studies will be necessary to prove the concept regarding downstream applications compared to other existing CRISPR platforms, especially in iPSC-derived neurons.

Major comments:

1. Quantification of CLTA-GFP silencing in HEK293T was evaluated using VLPs to deliver different epigenome editors. Were other types of carriers (e.g. liposomes or nanoparticles) used to compare the % CLTA-GFP to the RENDER-based CRISPR approach that this study proposed? This will provide a clear picture of why the development of VLP as a carrier of the CRISPR system is necessary and more effective than other published delivery forms when the same cell type is used for this comparison.
2. On the same note, how do the authors confirm the actual silencing vs. the simple lack of GFP signal? A secondary

validation method (e.g., methylation or histone acetylation levels, or mRNA measurements, especially for the V337M tau experiment) may be necessary.

3. Was cell viability tested while quantifying the percentages of gene silencing in Fig. 3a? If so, some normalization to the total cell count is needed.

4. In Fig. 4, the comparison between RENDER-CRISPRoff and mRNA electroporation in primary human T cells indicated that mRNA electroporation could silence the expression of CD55/CD8 significantly more than the RENDER approach. However, the cell viability with mRNA electroporation was lower. This highlights the need to clarify the threshold of % live versus % gene silencing.

5. Since the study showed some reduction of MAPT V337M in the cells using flow cytometry, isolating proteins from the cells and further investigating the changes in transcriptional regulation, and also the changes in the regulation of tauopathy-related genes and proteins would be interesting to show the proof of therapeutic concept as the authors proposed (see above point #2).

6. How was the neurite morphology of the iPSC-derived neuron after being treated with RENDER-CRISPRoff? Including images of neurons and biological processes in neurons after/before being treated with the RENDER-CRISPRoff will be helpful in clearly explaining how this platform works in the cells and how this gene-silencing treatment affects the changes in gene expression in the neural cells.

7. On a related note, the strategy can be tested in a mixed neuron/glia culture to demonstrate the cell-type-specific uptake and targeting of cargo-carrying VLPs. If not, at least some discussion of this approach in the mixed cell model (and eventually in vivo) should be included.

8. For the study of tau reduction through the mechanism of this RENDER-based CRISPR delivery, can authors elaborate on how this system mechanistically functions in editing the epigenome of the MAPT region rather than other neighboring genes or non-targeting genes in the cells, which possibly contribute to the tau reduction in neurons?

9. It may help to include a graphical abstract/schematic to show how exactly the RENDER-CRISPRoff works in gene silencing through epigenetic mechanisms. It can be either as a part of Figure 1 or a separate figure linking to the Discussion. Since this is a new technology, some kind of schematic would help readers picturize the exact mechanism of action of RENDER CRISPRoff system.

Minor comments:

1. The authors should elaborate on what types of cells mentioned in the introduction were challenging for genomic editing when the existing CRISPR platforms were used for gene silencing and/or gene editing.

2. In the discussion, it may be important to tie it back to existing CRISPR technology's limitations. How does this RENDER-based CRISPR technology solve the limitations of the CRISPR system for off-target delivery? Particularly in cell lines where the existing CRISPR platforms have not addressed this challenge. This point of view needs to be discussed.

Version 1:

Reviewer comments:

Reviewer #1

(Remarks to the Author)

The authors have addressed most of my previous concerns, and the revised manuscript represents a clear improvement over the original version. However, a few additional minor points arose upon reviewing the revised version. I encourage the authors to address the following comments to the best of their ability prior to final publication:

1. The authors initially adopted the published v4 BE-eVLP architecture [PMID: 35021064] and performed further stoichiometric optimization tailored to the RENDER platform. Since the development of the v4 BE-eVLP system, several studies have introduced additional engineering strategies to enhance gene editor packaging into eVLPs [PMID: 38191664; PMID: 39755699; PMID: 40209705]. Have the authors explored any of these more recent approaches to improve cargo (editor and sgRNA) loading in the RENDER system?

2. Lines 402–403: The authors should cite prior literature demonstrating that RNP delivery can reduce off-target editing.

3. In the iPSC-derived neuron experiment, while authors implied in the text that the plasmid transfection after neuronal differentiation is likely inefficient, it would strengthen the study to include a comparison of plasmid transfection alongside RENDER-mediated delivery in the iPSC-derived neuron as a proper control.

Reviewer #2

(Remarks to the Author)

The authors have performed careful and effective revisions. We regret that the TET1 data are not included in this revised version, but we support the acceptance of this work for publication.

Reviewer #3

(Remarks to the Author)

Reviewer #4

(Remarks to the Author)

The authors have addressed all prior concerns satisfactorily. No additional comments/suggestions noted. Good luck to all authors!

Dear Reviewers,

We sincerely thank you for your constructive feedback on our manuscript “Programmable epigenome editing by transient delivery of CRISPR epigenome editor ribonucleoproteins” (reference number: NCOMMS-24-79651-T). Below, we have provided detailed responses to the reviewers’ comments and outlined the revisions made to the manuscript.

Here you will find our responses colored in blue for each of your points.

Reviewer Comments and Responses

Reviewer #1:

In the present manuscript, the authors described the development of Robust Enveloped Delivery of Epigenome-editor Ribonucleoproteins (RENDER), that delivers epigenetic editors as RNP by packaging and delivering epigenetic editor cargoes in the VLPs. The authors optimized the RENDER for improved cargo packaging, and demonstrated RENDER-mediated epigenetic editing in primary T cells and iPSC-derived neurons. The effort to deliver epigenetic editors transiently as RNP cargo is highly commendable. Additional demonstration of the reversibility of epigenome editing, and multiplexability of RENDER platform also adds significant value to the study.

The manuscript is generally well-written, with concise and accurate language. The figures and data support the conclusions effectively, and the experiments conducted are comprehensive and well-reasoned. I recommend the manuscript for publication if the authors can sufficiently address the following comments.

We thank the Reviewer for their positive assessment of our work.

Comment 1:

Primary T cell targeting – Have the authors tested multiple doses for T-cell transduction? The Methods section indicates that only 5 μ L of 100 \times concentrated RENDER was used for T-cell transduction, which appears to be a very low dose for a hard-to-transduce cell type. Was this dose chosen due to concerns about toxicity? If possible, it would be valuable for the authors to test multiple doses and provide a dose-response curve. This would also allow for a comparison of editing efficiency with mRNA electroporation, enhancing the comprehensiveness of the study.

Response:

We appreciate the reviewer’s insightful comment. Initially we chose to transduce T cells with 5 μ L RENDER, following established protocols for virus-like particle transduction in T cells (Banskota et al., Cell 2022). We have now performed additional experiments testing multiple doses of 100 \times concentrated RENDER for T cell transduction. Specifically, we prepared RENDER-CRISPRoff targeting *CD55* or *CD81*, and transduced primary human T cells isolated from PBMCs with different doses ranging from 1 μ L to 16 μ L. Flow cytometry and antibody staining results show that, within this dose range, the proportion of cells with *CD55* or *CD81* silenced increases as the RENDER dose increases, indicating a positive correlation between silencing efficiency and dose. Dose-response silencing efficiency bar plots are now included in Fig. 4f and 4g.

Additionally, cell viability bar plots are now included in Extended Data Fig. 10b, 10c and 10d.

These results are discussed in the revised manuscript.

Comment 2:

eVLP production method – The manuscript describes collecting VLP supernatant three days post-transfection, while many existing studies collect VLPs 40–48 hours post-transfection. Have the authors performed a side-by-side comparison of these collection times? If not, could the authors clarify the rationale for this deviation from standard protocols?

Response:

We thank the reviewer for pointing this out. In our initial experiments, we found that collecting VLPs at 72 hours post-transfection resulted in a significantly higher yield compared to collecting at 48 hours. To further clarify these results, we performed a side-by-side comparison of VLP collection at both 48- and 72-hours post-transfection. Specifically, we treated CLTA-GFP HEK293T reporter cells with VLPs collected at these two time points and assessed gene silencing efficiency five days post-treatment. These results show that cells treated with VLPs collected at 72 hours had a higher percentage of gene silencing compared to those treated with VLPs collected at 48 hours, under the same volume conditions.

Additionally, we quantified the amount of Cas9 (*S. pyogenes*) protein in both concentrated VLP samples using ELISA. Since CRISPRoff is a fusion protein that contains dCas9, we used Cas9 protein levels as a measure of the CRISPRoff editor protein concentration. The results show that the CRISPRoff protein concentration in the 48-hour sample was significantly lower than that in the 72-hour sample. These data indicate that collecting VLPs at 72 hours post-transfection significantly improves the VLP yield and thereby improving the silencing efficiency. Quantification plot of spCas9 protein in the concentrated eVLP samples harvested at 48 hours and 72 hours post-transfection is now included in the **Fig. 1c**. These results are discussed in the revised manuscript. We also updated the Methods section to include details of the ELISA experiments.

Comment 3:

Line 163-165 – The authors conclude that lower doses of RENDER-mediated delivery of CRISPRoff result in reduced durability. This observation is rather intriguing, as durability would typically depend on the editor itself rather than the delivery method or dose. Can the authors speculate on the underlying mechanism for this phenomenon?

Response:

We thank the reviewer for highlighting this intriguing observation. Stable and durable gene silencing requires sufficient levels of the CRISPRoff editor to enter the cell and induce adequate

DNA methylation. At lower doses of RENDER-mediated delivery, the amount of editor delivered to each cell may be insufficient to achieve the threshold level of epigenetic modifications necessary for stable, heritable gene silencing. This limitation could explain the observed reduction in durability at lower doses.

Previous studies support this idea by demonstrating a dose-dependent correlation between changes in target gene expression and the induction level of the epigenetic editing machinery (Neumann et al., *Science* 2025; Policarpi et al., *Nature Genetics* 2024; Policarpi et al., *BioEssays* 2021). However, the specific threshold of epigenetic modifications required for stable gene silencing is still under investigation and is likely gene-dependent.

Comment 4:

The study demonstrates efficient editing with RENDER in cell culture. To increase the significance of the manuscript, it is highly recommended that the authors explore the *in vivo* translatability of the platform. Demonstrating RENDER-mediated delivery of epigenetic editors in mouse models would add substantial impact to the study.

Response:

We appreciate the reviewer's suggestion to explore the *in vivo* translatability of RENDER, which represents a valuable direction for future research. While our study primarily focuses on developing and optimizing methods for delivering epigenome editors and evaluating the reliability of this approach for RNP-based epigenome editing, we fully recognize the potential impact of demonstrating RENDER-mediated delivery *in vivo*. Recent studies from other laboratories (Banskota et al., *Cell* 2021; Hamilton et al., *Nature Biotechnology* 2024; An et al., *Nature Biotechnology* 2024; Geilenkeuser et al., *Cell* 2025) have shown promising results with VLP-based *in vivo* delivery of Cas nucleases, base editors and prime editors, highlighting its potential as a powerful tool for *in vivo* gene editing. We have now included a discussion on the future potential of RENDER for *in vivo* epigenome editing in the revised manuscript:

“Our data showed that RENDER-based delivery of CRISPRoff results in robust silencing of endogenous genes in cultured human neurons, including notable repression of Tau protein in neurons harboring a known pathogenic mutation. These results highlight the therapeutic potential of RENDER-CRISPRoff as a strategy for treating neurodegenerative diseases. Recent studies have shown the promise of virus-like particles (VLPs) as vehicles for *in vivo* gene editor RNP delivery, and the RENDER platform extends this approach to epigenome editors, laying the groundwork for future *in vivo* epigenetic therapies. Furthermore, recent advancements in envelope protein engineering have further shown the feasibility of reprogramming virus/virus-like particle targeting tropism to specific cell types in complex environments, including *in vivo*, supporting the potential of RENDER for tissue-specific delivery and epigenome editing. Beyond advances in epigenome editor RNP delivery, RENDER's high efficacy in CRISPRoff apoprotein delivery can offer a potential platform for CRISPR screens in sgRNA library-expressing cell lines, facilitating functional genomics applications through epigenome editing perturbations. Thus, RENDER expands the potential of programmable epigenome editors for both biomedical research and therapeutic applications.”

Reviewer #2

In their article, “Programmable Epigenome Editing by Transient Delivery of CRISPR Epigenome Editor Ribonucleoproteins”, Da Xu et al. propose and extensively demonstrate how their RENDER platform efficiently delivers CRISPR epigenome editors to various human cells, including primary T cells and neuronal cells, which are typically difficult to transfect. The study is experimentally robust and of considerable scientific interest. However, there are several points that require further clarification.

We thank the Reviewer for their positive assessment of our work.

Comment 1:

Figure 1: The authors primarily focus on inducing gene downregulation using an endogenously GFP-tagged gene as their target. Only a single experiment was conducted using an editor enzyme responsible for increasing gene expression, TET (Fig. 1i-j). The efficiency of gene activation demonstrated is significantly lower than that of downregulation. The authors state: “Further optimization experiments are needed to address this issue” (lines 154-157). However, the authors should provide a more detailed explanation for the low activity of this construct as in their previous study (doi: 10.1016/j.cell.2021.03.025), plasmid transfection achieved stable reactivation in 74% of the cells. The authors should elaborate on the advantages of using the RENDER platform despite the reduced efficiency compared to plasmid transfection. Additionally, what percentage of activation would be achieved if another endogenous gene was targeted?

Response:

Thank you for your insightful comment. In response to your suggestion, we have further explored epigenome gene activator delivery using the RENDER platform. As noted in our previous report (Nuñez et al., Cell 2021), we know that the orientation of the epigenetic effector domain (e.g., the KRAB domain) relative to dCas9 affects editing efficiency. Therefore, we investigated the impact of TET1 and dCas9 orientation on delivery and editing efficiency. We constructed a dCas9-TET1(CD) fusion protein and compared the ability of TET1(CD)-dCas9 and dCas9-TET1(CD) to activate silenced genes within the RENDER delivery system. Our results show that TET1(CD)-dCas9 induced detectable reactivation of *CLTA*, with efficiency positively correlated to the dose, while dCas9-TET1(CD) only showed very weak reactivation, suggesting that the architecture of the TET1(CD) editor protein affects its functionality.

Following the reviewer’s suggestion, we also tested reactivating another endogenous gene, the silenced *H2B* gene, but no significant reactivation was observed for either TET1(CD) editor using RENDER delivery.

Additionally, we quantified the amount of Cas9 protein in these concentrated RENDER samples using ELISA to assess VLP packaging efficiency. The results showed that the amount of editor in both TET1(CD)-dCas9 and dCas9-TET1(CD) RENDER samples was lower than in RENDER-CRISPRoff v1, indicating that the nature of the editor protein affects VLP packaging.

We speculate that plasmid transfection (Nuñez et al., Cell 2021), which produces high concentrations of the editor over several days, may contribute to the higher efficiency observed in that system. Furthermore, previous work with eVLP-based editors and eVLP-prime editors (Banskota et al., Cell 2021; An et al., Nature Biotechnology 2024) highlighted that the properties and mechanisms of the editor itself significantly impact VLP-based delivery and editing efficiency. Based on these observations, we hypothesize that further optimization of the RENDER system, particularly with regards to packaging and delivery for different editors like TET1(CD)-dCas9 (~7.4 kb), which is even larger than CRISPRoff (~7.0 kb), will be required to improve activation efficiency.

Comment 2:

Figure 3a: The authors report low silencing efficiency in K562 cells, attributing this to cell type limitations (lines 206-207). Did they attempt to assess differential uptake or to optimize delivery conditions, for example, by increasing the doses of the editor? Such experiments would strengthen their conclusions.

Response:

Thank you for your valuable suggestion. Based on previous report (Nuñez et al., Cell 2021, Figure S2B), we observed that dox-inducible expression of CRISPRoff in K562 cells can achieve high levels of *CD81* silencing, indicating that CRISPRoff can effectively silence *CD81* in K562 cells when delivered at high levels. Therefore, we suspect that the low silencing efficiency observed in Fig. 3a in this manuscript may be due to suboptimal delivery of RENDER into K562 cells.

In response to your suggestion, we tested different doses of RENDER-CRISPRoff (ranging from 0.5 to 32 μ l) in K562 cells. Our results showed that both *CD55* and *CD81* silencing levels correlated positively with the RENDER dose, and increasing the dose improved gene silencing efficiency in K562 cells. Dose-response silencing efficiency curve plot are now included in **Extended Data Fig. 4a**.

Additionally, we explored the use of viral transduction enhancers, polybrene and retronectin, in combination with our RENDER delivery platform in K562 cells. Both transduction enhancers, widely used to enhance viral transduction, were effective in increasing the silencing efficiency of RENDER-CRISPRoff in K562. Quantification of *CD55* and *CD81* silencing in K562 cells bar plots are now included in **Extended Data Fig. 4b** and **4c**.

Additionally, we have updated these results in the manuscript.

Comment 3:

Figure 3b: The authors present genome-wide methylation data to demonstrate the high specificity of the approach. However, they state (lines 218-221) that “modest” methylation events unrelated to the *CD55* target were detected. It would be important to determine whether these slight off-target methylations are associated with biological effects. The authors should include gene expression data to assess the potential functional impact of these off-target effects.

Response:

We thank the Reviewer for raising this critical point. To address this, we performed additional experiments to evaluate the potential off-target effects of RENDER-CRISPRoff by measuring global changes in gene expression. Specifically, we performed RENDER-CRISPRoff (targeting *CD55*) treatment on Jurkat cells following the procedure we did for the genome-wide EM-seq profiling experiment, and then performed RNA sequencing (RNA-seq) to assess transcriptomic changes in RENDER-CRISPRoff treated cells, with untreated cells (UT) as controls. Consistent with EM-seq results, the target gene *CD55* was the most significantly downregulated gene in cells treated with RENDER-CRISPRoff v3 targeting *CD55*, supporting the on-target epigenome editing specificity of this approach. No other genes, except for *APOM* (*ENSG00000235754.7*), which is relatively lowly expressed in both treated and untreated cells, showed significant changes after treatment (adjusted $P < 10^{-5}$, $|\log_2FC| > 2$).

Additionally, we performed RNA-seq on RENDER-CRISPRoff-treated and untreated HEK293T cells. We found that, consistent with previously reported DNA methylation profiles of CRISPRoff-edited HEK293T cells (Nuñez et al., Cell 2021), the target gene *CLTA* was the most significantly downregulated. One gene, *TIMM22* (*ENSG00000177370.5*), with low expression (<1 transcripts per million (TPM)), showed upregulation after treatment (adjusted $P < 10^{-5}$, $|\log_2\text{FC}| > 2$).

Combined with genome-wide EM-seq data we present in this manuscript, as well as previous work (Nuñez et al., Cell 2021), these results showed efficient on-target activity, evidenced by increased DNA methylation at the target regions and a corresponding decrease in transcript levels. The most significant and profound changes in mRNA expression were specific to the targeted genes, *CD55* and *CLTA*. Together, our results support the high specificity of RENDER-CRISPRoff-mediated gene silencing with minimal off-target effects.

In response to the reviewer's comment, we have revised the relevant sentences, the associated figures (Fig. 3d, 3e and Extended Data Fig. 8) and figure legends to provide a clearer and more detailed explanation.

Comment 4:

Figure 4: The authors compare the efficiency of editing via the RENDER platform to that of mRNA nucleofection. They highlight the RENDER platform's advantage in maintaining greater cell viability compared to mRNA transfection via electroporation (lines 67-69, 347-349). While higher cell viability is a benefit, the data in Fig. 4d-e show that mRNA transfection results in significantly higher silencing of CD55 cells at both 2 days (43.7%) and 11 days (97%), compared to 24% and 21% with RENDER. The authors should emphasize why the RENDER platform should be preferred over mRNA, perhaps by discussing other advantages such as cost or preparation time.

Response:

We thank the reviewer for the valuable comment. To address this, we have now performed additional experiments testing multiple doses of 100× concentrated RENDER for primary human T-cell transduction to better explore the editing efficiency of RENDER-CRISPRoff in primary human T cells. Specifically, we prepared RENDER-CRISPRoff targeting *CD55* or *CD81* and transduced primary human T cells isolated from PBMCs with different doses (ranging from 1 μ l to 16 μ l). Flow cytometry and antibody staining results show that, within this dose range, the proportion of cells with *CD55* or *CD81* silenced increases as the RENDER dose increases (up to ~45% for *CD55*, and ~60% for *CD81*), and no significant cell death was observed within this dose range.

On the other hand, mRNA electroporation causes significant cell death, so the effective editing percentage in practice is a product of the total number of viable cells and the effective editing rate. For example, with mRNA electroporation, the effective editing percentage is $55\% \times 97\% = 53\%$, meaning 53% of primary human T cells from PBMCs are edited. In contrast, with RENDER-CRISPRoff, the effective editing percentage is $100\% \times 45\% = 45\%$. Therefore, when considering the total number of edited cells, the editing efficiency of RENDER-CRISPRoff is comparable to that of mRNA electroporation.

A dose-response silencing efficiency curve is now included in **Fig. 4f** and **4g**, as well as a dose-response cell viability bar plots in **Extended Data Fig. 10b, 10c** and **10d**. These results are discussed in the revised manuscript.

Furthermore, if the silencing efficiency of RENDER for endogenous genes such as CD55 and CD81 is between 20% and 35%, the authors should clarify its advantage over plasmid DNA. They note (lines 644-646) that plasmid DNA requires cell selection methods, such as sorting or enzymatic selection, due to low transfection and silencing efficiency. However, as seen in Fig. 4d and 4f, RENDER's silencing efficiency also appears low and may similarly require additional cell selection steps, depending on the gene. This point should be addressed in the discussion.

Response:

We thank the Reviewer for raising this important point, and we apologize for the confusion. Our statement in lines 344-347, "In comparison, plasmid DNA transfections of CRISPRi/off into the same cells yield ~60% transfection efficiencies, necessitating the use of selection methods (e.g., cell sorting or antibiotic selection) to isolate a pure population of transfected cells," was intended to highlight that RENDER-CRISPRoff achieves high delivery and editing efficiencies in HEK293T, Jurkat, and iPSC-derived neurons, surpassing the limitations of traditional plasmid DNA transfections, which typically require additional selection methods (e.g., cell sorting or antibiotic selection) due to their lower transfection efficiencies. This makes RENDER a more efficient tool, reducing the need for additional steps in applications such as CRISPR screens.

To address the reviewer's concern, we have revised the relevant section of the discussion to more clearly and accurately describe the advantages of RENDER-CRISPRoff over plasmid DNA transfection in terms of its efficiency and ease of use in various cell types.

Comment 5:

Extended Data Figs 4 and 5: The statement that methylation was "gained" in untreated cells appears unclear. The authors are advised to revise this sentence and provide a more detailed explanation of the figures.

Response:

We thank the Reviewer for this suggestion. To address this, we have revised the relevant sentences, the associated figures (**Extended Data Fig. 6** and **7**) and figure legends to provide a clearer and more detailed explanation.

Minor comments:

Paragraph “Multiplexed Epigenome Editing...” (line 244): The use of terms such as “multiplexed” and “multiple” to describe targeting of only 2 genes seems overstated. The authors might rephrase this section for accuracy.

Response:

We thank the Reviewer for this suggestion. In the revised manuscript we have rephrased this paragraph, primarily changing “multiplexed” to “combinatorial”.

Line 255-258: The concept of the apo-form should be explained more clearly.

Response:

We thank the Reviewer for pointing this out. To clarify, we have revised the text to:

“CRISPR screens leverage the efficiency and flexibility of CRISPR–Cas genome editing, making them a powerful tool for biological discovery. In a typical pooled CRISPR screen, a gRNA library is introduced into cells, with each cell receiving a different gRNA, typically delivered via lentiviral transduction. The CRISPR–Cas protein is introduced either stably or transiently. We hypothesized that RENDER could provide a novel approach for transiently delivering epigenome editors in their apo-form (sgRNA-unbound proteins), which then assemble with sgRNAs inside the target cell to form functional RNP complexes, thereby expanding the methods of pooled CRISPR screens.”

Materials and Methods: The article does not provide details on how mRNA was obtained or the conditions used for the electroporation reaction. These details should be included.

Response:

The CRISPRoff mRNA was synthesized and purchased from Aldeveron, as described in the previous manuscript (line 486). The mRNA electroporation experiment in primary human T cells was performed as previously reported (Pattali et al., bioRxiv 2024), where detailed information on the procedure was provided. This reference was cited in the previous manuscript (line 487). Since the paper was recently published in *Methods in Enzymology*, we have updated the citation to (Pattali et al., *Methods in Enzymology* 2025).

Terminology Consistency: Throughout the article, the term “RENDER-CRISPRoff v3” should be used consistently wherever these conditions are applied (e.g., line 200).

Response:

We have revised the manuscript to ensure that the term “RENDER-CRISPRoff v3” is used consistently throughout the article.

Reviewer #3

We thank the Reviewer for their positive assessment of our work.

Reviewer #4

The manuscript by Xu D. et al. developed viral-like particles (VLPs) to deliver epigenome-editor RNPs named RENDER. The main focus was on quantitative assessments of gene silencing in several types of human cell lines using the RENDER-CRISPRoff platform. This platform was used to knock out CD55/CD81 in primary T cells and also to silence CD81 and tau expression through targeted MAPT mutation at V337M in iPSC-derived neurons. However, more studies will be necessary to prove the concept regarding downstream applications compared to other existing CRISPR platforms, especially in iPSC-derived neurons.

Major comments:

1. Quantification of CLTA-GFP silencing in HEK293T was evaluated using VLPs to deliver different epigenome editors. Were other types of carriers (e.g. liposomes or nanoparticles) used to compare the % CLTA-GFP to the RENDER-based CRISPR approach that this study proposed? This will provide a clear picture of why the development of VLP as a carrier of the CRISPR system is necessary and more effective than other published delivery forms when the same cell type is used for this comparison.

Response:

We thank the Reviewer for this comment. Due to the nature of epigenome editors, such as CRISPRoff, which consist of chromatin-modifying domains that are often difficult to purify at high concentrations in vitro for RNP delivery, we were unable to assemble them as RNPs into carriers such as liposomes or nanoparticles for a direct comparison with RENDER.

In our previous studies (Nuñez et al., 2021; Pattali et al., 2025), we performed plasmid transfection to deliver CRISPRoff-encoding plasmids to CLTA-GFP HEK293T cells. The results revealed that transfection efficiency was typically around 60%, meaning only 60% of the transfected cells expressed CRISPRoff and underwent epigenome editing. In contrast, using RENDER-CRISPRoff, we achieved stable repression of GFP in 92% of treated cells at 49 days post-treatment, as shown in Fig. 2g and 2h, demonstrating that RENDER-CRISPRoff is more efficient than plasmid transfection in the same cell line.

Additionally, our findings align with those in Banskota et al. (2022, Cell), where VLP-delivered base editors showed comparable or higher on-target editing efficiency compared to plasmid transfection in HEK293T cells (Figure 3G). Similarly, in Ramadoss et al. (2024, bioRxiv), VLP delivery of Cas9 RNPs in iPSC-derived neurons achieved up to 60% indel efficiency (Fig. 2b), whereas LNP-delivered Cas9 mRNA only resulted in less than 30% indels (Fig. 4b).

Overall, our results, along with those from other studies, show that VLP-based delivery offers comparable or even higher efficiency than plasmid transfection and mRNA LNP delivery.

2. On the same note, how do the authors confirm the actual silencing vs. the simple lack of GFP signal? A secondary validation method (e.g., methylation or histone acetylation levels, or mRNA measurements, especially for the V337M tau experiment) may be necessary.

Response:

We thank the Reviewer for raising this critical point. Regarding CLTA-GFP silencing, the CLTA-GFP HEK293T reporter line used in our study comes from previous work (Leonetti et al., 2016). This cell line has the advantage of GFP directly tagging the endogenous *CLTA* gene without altering the promoter or gene body, and it has been widely used in epigenome editing research for efficient and convenient readout of gene silencing. For example, in Nuñez et al. (Cell 2021), RNA-seq of GFP-negative cells treated with CRISPRoff showed a significant reduction in *CLTA* mRNA levels, demonstrating that GFP expression closely correlates with CLTA expression.

To further validate the reliability of this reporter system and rule out any potential changes due to differences in delivery methods (from plasmid transfection in previous work to RENDER), and to better assess the specificity of RENDER-CRISPRoff, we performed additional experiments measuring global changes in gene expression. Due to time and cost constraints, we chose two different cell types: CLTA-GFP HEK293T and Jurkat cells. Specifically, we first performed RENDER-CRISPRoff (targeting *CD55*) treatment on Jurkat cells following the procedure we did for the genome-wide EM-seq profiling experiment, and then performed RNA sequencing (RNA-seq) to assess transcriptomic changes in RENDER-CRISPRoff treated cells, with untreated cells (UT) as controls. Consistent with EM-seq results, the target gene *CD55* was the most significantly downregulated gene in cells treated with RENDER-CRISPRoff v3 targeting *CD55*, supporting the on-target epigenome editing specificity of this approach. No other genes, except for *APOM* (*ENSG00000235754.7*), which is relatively lowly expressed in both treated and untreated cells, showed significant changes after treatment (adjusted $P < 10^{-5}$, $|\log_2FC| > 2$).

Additionally, we performed RNA-seq on RENDER-CRISPRoff-treated and untreated HEK293T cells. We found that, consistent with previously reported DNA methylation profiles of CRISPRoff-edited HEK293T cells (Nuñez et al., Cell 2021), the target gene *CLTA* was the most significantly downregulated. One gene, *TIMM22* (*ENSG00000177370.5*), with low expression (<1 transcripts per million (TPM)), showed upregulation after treatment (adjusted $P < 10^{-5}$, $|\log_2FC| > 2$).

Combined with genome-wide EM-seq data we present in this manuscript, as well as previous work (Nuñez et al., Cell 2021), these results showed efficient on-target activity, evidenced by increased DNA methylation at the target regions and a corresponding decrease in transcript levels. The most significant and profound changes in mRNA expression were specific to the targeted genes, *CD55* and *CLTA*. Together, our results support the high specificity of RENDER-CRISPRoff-mediated gene silencing with minimal off-target effects.

In response to the reviewer's comment, we have revised the relevant sentences, the associated figures (**Fig. 3d, 3e and Extended Data Fig. 8**) and figure legends to provide a clearer and more detailed explanation.

3. Was cell viability tested while quantifying the percentages of gene silencing in Fig. 3a? If so, some normalization to the total cell count is needed.

Response:

We thank the Reviewer for the comment. In the experiment presented in Fig. 3a, cell viability was not assessed using viability staining. However, compared to the untreated control group, no noticeable cell death was observed in the RENDER-treated group. While in Fig. 4b and 4c, we used Zombie Aqua (viability dye) staining to assess cell viability of T cells. In response to the reviewer's suggestion, we have now revised **Fig. 4c** by normalizing the data to the untreated control group for better clarity.

4. In Fig. 4, the comparison between RENDER-CRISPRoff and mRNA electroporation in primary human T cells indicated that mRNA electroporation could silence the expression of CD55/CD8 significantly more than the RENDER approach. However, the cell viability with mRNA electroporation was lower. This highlights the need to clarify the threshold of % live versus % gene silencing.

Response:

We thank the Reviewer for the comment. As mentioned in our response to Comment #3, we have now revised **Fig. 4c** by normalizing the data to the untreated control group for better clarity.

To further address this point, we performed additional experiments testing multiple doses of 100× concentrated RENDER for primary human T-cell transduction to better explore the editing efficiency of RENDER-CRISPRoff in primary human T cells. Specifically, we prepared RENDER-CRISPRoff targeting *CD55* and *CD81* and transduced primary human T cells isolated from PBMCs with different doses (ranging from 1 μl to 16 μl). Flow cytometry and antibody staining results showed that, within this dose range, the proportion of cells with downregulated CD55 and CD81 expression increased as the RENDER dose increased (up to ~45% for *CD55* and ~60% for *CD81*), with no significant cell death observed.

A dose-response silencing efficiency curve is now included in **Fig. 4f** and **4g**, as well as a dose-response cell viability bar plots in **Extended Data Fig. 10b**, **10c** and **10d**. These results are discussed in the revised manuscript.

On the other hand, mRNA electroporation causes significant cell death, so the effective editing percentage in practice is a product of the total number of viable cells and the effective editing rate. For example, with mRNA electroporation, the effective editing percentage is $55\% \times 97\% = 53\%$, meaning 53% of primary human T cells from PBMCs are edited. In contrast, with RENDER-CRISPRoff, the effective editing percentage is $100\% \times 45\% = 45\%$. Therefore, when considering the total number of edited cells, the editing efficiency of RENDER-CRISPRoff is comparable to that of mRNA electroporation. These results are discussed in the revised manuscript.

5. Since the study showed some reduction of MAPT V337M in the cells using flow cytometry, isolating proteins from the cells and further investigating the changes in transcriptional regulation,

and also the changes in the regulation of tauopathy-related genes and proteins would be interesting to show the proof of therapeutic concept as the authors proposed (see above point #2).

Response:

We concur with the Reviewer’s suggestion that investigating the changes in the regulation of tauopathy-related genes and proteins would be an interesting direction for future research. As mentioned in our response to Comment #2, due to time and cost constraints, we were unable to perform protein level, DNA methylation, and mRNA analyses for all cell types tested with RENDER-CRISPRoff. However, the new data from our additional experiments support that “actual silencing” occurred in different cell types, and RENDER-CRISPRoff demonstrated high specificity.

The focus of this study was to introduce RENDER as a solution for delivering epigenome editors, to test the applicability of RENDER, and to validate the specificity of CRISPRoff-based editing using RENDER. We believe the current data provide proof of concept that RENDER-CRISPRoff holds potential as a therapeutic tool. In future research, we aim to perform more detailed studies on epigenome editing targeting disease-related genes, including Tau, laying the foundation for therapeutic epigenome editing.

6. How was the neurite morphology of the iPSC-derived neuron after being treated with RENDER-CRISPRoff? Including images of neurons and biological processes in neurons after/before being treated with the RENDER-CRISPRoff will be helpful in clearly explaining how this platform works in the cells and how this gene-silencing treatment affects the changes in gene expression in the neural cells.

Response:

We thank the Reviewer for the comment. Regarding the morphology of iPSC-derived neurons after RENDER-CRISPRoff treatment, we took images of neurons treated with RENDER-CRISPRoff and untreated neurons at 7 days post-treatment.

To better answer the reviewer's question, we recommend the work by Ramadoss et al. (2024, bioRxiv), in which the authors used VLP delivery of Cas9 in iPSC-derived neurons. This study,

which shares similar cell types and experimental methods with our manuscript, did not observe significant changes in neuron morphology after treatment with VLP, which aligns with our findings. These results suggest that VLP treatment does not have a significant effect on the morphology of iPSC-derived neurons.

Furthermore, in our previous studies, iPSC-derived neurons treated with CRISPRoff transfection did not show noticeable morphological differences compared to untreated iPSC-derived neurons, indicating that CRISPRoff-induced gene silencing also had no significant effect on neuron morphology.

7. On a related note, the strategy can be tested in a mixed neuron/glial culture to demonstrate the cell-type-specific uptake and targeting of cargo-carrying VLPs. If not, at least some discussion of this approach in the mixed cell model (and eventually *in vivo*) should be included.

Response:

We concur with the Reviewer's suggestion that investigating cell-type-specific targeting of cargo-carrying VLPs would be important for future applications, including *in vivo* targeted delivery. In response to this, we referenced two recent studies: Hamilton et al., 2024 and Strebinger et al., 2023, which highlight the feasibility of modulating virus/virus-like particle targeting specificity through envelope protein engineering. We have revised the discussion to include this future direction:

“Our data showed that RENDER-based delivery of CRISPRoff results in robust silencing of endogenous genes in cultured human neurons, including notable repression of Tau protein in neurons harboring a known pathogenic mutation. These results highlight the therapeutic potential of RENDER-CRISPRoff as a strategy for treating neurodegenerative diseases. Recent studies have shown the promise of virus-like particles (VLPs) as vehicles for *in vivo* gene editor RNP delivery, and the RENDER platform extends this approach to epigenome editors, laying the groundwork for future *in vivo* epigenetic therapies. Furthermore, recent advancements in envelope protein engineering have further shown the feasibility of reprogramming virus/virus-like particle targeting tropism to specific cell types in complex environments, including *in vivo*, supporting the potential of RENDER for tissue-specific delivery and epigenome editing. Beyond advances in epigenome editor RNP delivery, RENDER's high efficacy in CRISPRoff apoprotein delivery can offer a potential platform for CRISPR screens in sgRNA library-expressing cell lines, facilitating functional genomics applications through epigenome editing perturbations. Thus, RENDER expands the potential of programmable epigenome editors for both biomedical research and therapeutic applications.”

8. For the study of tau reduction through the mechanism of this RENDER-based CRISPR delivery, can authors elaborate on how this system mechanistically functions in editing the epigenome of the MAPT region rather than other neighboring genes or non-targeting genes in the cells, which possibly contribute to the tau reduction in neurons?

Response:

In the RENDER-CRISPRoff *MAPT* silencing experiment, we used sgRNAs specifically designed for *MAPT* silencing (Nuñez et al., Cell 2021). In unpublished data, we experimentally derived the top sgRNAs for CRISPRi in iPSC-derived neurons targeting MAPT-WT or MAPT-V337M. Those

were the sgRNAs that we chose for the experiments in our manuscript. As depicted in the plot, the top sgRNAs are centered at the TSS of *MAPT*. The green bar is the annotated CpG island for *MAPT*. Guided by the sgRNA, the CRISPRoff RNP complex precisely targets the promoter region of the *MAPT* gene, leading to gene silencing through the deposition of DNA methylation, which is catalyzed by the DNMT3A domain in CRISPRoff. The specificity of this sgRNA targeting has been validated in previous studies. In Nuñez et al. (2021), we profiled DNA methylation and determined that CRISPRoff-mediated DNA methylation was specifically deposited within a 1.5–2 kb window surrounding the targeted site (Figure 2F and 2G), demonstrating the high specificity of CRISPRoff-mediated epigenome editing.

The human *MAPT* gene is a large (~134 kb) gene located on chromosome 17q21 (NC_000017.11 (45894554..46028334)), with over 10 kb of flanking sequence on both sides lacking any annotated genes. Therefore, it is highly unlikely that CRISPRoff would inadvertently target “other neighboring genes or non-targeting genes,” which could contribute to Tau reduction in neurons.

9. It may help to include a graphical abstract/schematic to show how exactly the RENDER-CRISPRoff works in gene silencing through epigenetic mechanisms. It can be either as a part of Figure 1 or a separate figure linking to the Discussion. Since this is a new technology, some kind of schematic would help readers picturize the exact mechanism of action of RENDER CRISPRoff system.

Response:

We thank the Reviewer for the suggestion. In response, we have added a graphical schematic in **Extended Data Fig. 11**, illustrating how RENDER delivers CRISPRoff into cells and how CRISPRoff induces gene silencing through epigenetic mechanisms. The figure legend provides a detailed description of the process.

“The envelope glycoprotein protein on the VLP surface facilitates fusion between the VLP and host cell membranes. Within the cytoplasm, the mature core packaging editor RNPs dissociate, and RNPs are released. The nuclear localization signal (NLS) on the editor proteins guides the RNPs into the nucleus. Once in the nucleus, the RNPs recognize and bind to the target DNA sequence, leading to the deposition of DNA methylation and histone modifications, which ultimately represses target gene expression.”

Minor comments:

1. The authors should elaborate on what types of cells mentioned in the introduction were challenging for genomic editing when the existing CRISPR platforms were used for gene silencing and/or gene editing.

Response:

We appreciate the Reviewer’s comments and recognition that RENDER broadens the applicability of epigenome editing across a range of cell types. In this work, we demonstrate the effectiveness of RENDER-CRISPRoff in achieving epigenome editing across eight different cell lines, as well as in primary human T cells and iPSC-derived neurons. Although previous studies have utilized plasmid transfection, mRNA nucleofection, or mRNA-LNPs to deliver CRISPR gene editors to the aforementioned cell types, none of these methods offer a convenient, versatile, low-toxicity delivery solution that is not heavily dependent on specialized equipment. RENDER simplifies the delivery process across various cell types, significantly reducing these constraints. This advancement enhances the versatility of CRISPRoff and potentially many other epigenome editing tools for both research and clinical applications.

In response to the reviewer’s suggestion, we have revised the discussion accordingly.

“Through our experiments, we have demonstrated the robust applicability and potency of RENDER across various cell types, addressing prior challenges in delivery while further improving editing efficiency. In the laboratory, plasmid DNA transfections or lentiviral infection of CRISPRi/off typically yield suboptimal transfection efficiencies, often necessitating additional selection methods (e.g., cell sorting or antibiotic selection) to isolate a pure population of editor-carrying cells. Additionally, mRNA electroporation is highly dependent on reagents and equipment, and inevitably reduces cell viability. Furthermore, these methods are often not suitable for key cell types, such as neurons. While CRISPRoff and other epigenome editors have been delivered as DNA in AAV vectors or mRNA in LNPs recently, to our knowledge, our work is the first to report direct RNP delivery of epigenome editors. RNP delivery offers several advantages, including its transient nature and minimized off-target editing. Our results demonstrate that RENDER effectively addresses these challenges, providing a more efficient solution for epigenome editor delivery.”

2. In the discussion, it may be important to tie it back to existing CRISPR technology's limitations. How does this RENDER-based CRISPR technology solve the limitations of the CRISPR system for off-target delivery? Particularly in cell lines where the existing CRISPR platforms have not addressed this challenge. This point of view needs to be discussed.

Response:

We thank the Reviewer for the suggestion. In response, we have expanded the discussion to further highlight the advantages of the RENDER platform.

“Through our experiments, we have demonstrated the robust applicability and potency of RENDER across various cell types, addressing prior challenges in delivery while further improving editing efficiency. In the laboratory, plasmid DNA transfections or lentiviral infection of CRISPRi/off typically yield suboptimal transfection efficiencies, often necessitating additional selection methods (e.g., cell sorting or antibiotic selection) to isolate a pure population of editor-carrying cells. Additionally, mRNA electroporation is highly dependent on reagents and equipment, and inevitably reduces cell viability. Furthermore, these methods are often not suitable for key cell types, such as neurons. While CRISPRoff and other epigenome editors have been delivered as DNA in AAV vectors or mRNA in LNPs recently, to our knowledge, our work is the first to report direct RNP delivery of epigenome editors. RNP delivery offers several advantages, including its transient nature and minimized off-target editing. Our results demonstrate that RENDER effectively addresses these challenges, providing a more efficient solution for epigenome editor delivery.”

Dear Reviewers,

We sincerely thank you for your constructive feedback on our manuscript “Programmable epigenome editing by transient delivery of CRISPR epigenome editor ribonucleoproteins” (reference number: NCOMMS-24-79651-T). Below, we have provided detailed responses to the reviewers’ comments and outlined the revisions made to the manuscript.

Here you will find our responses colored in blue for each of your points.

Reviewer Comments and Responses

Reviewer #1 (Remarks to the Author):

The authors have addressed most of my previous concerns, and the revised manuscript represents a clear improvement over the original version. However, a few additional minor points arose upon reviewing the revised version. I encourage the authors to address the following comments to the best of their ability prior to final publication:

Comment 1:

The authors initially adopted the published v4 BE-eVLP architecture [PMID: 35021064] and performed further stoichiometric optimization tailored to the RENDER platform. Since the development of the v4 BE-eVLP system, several studies have introduced additional engineering strategies to enhance gene editor packaging into eVLPs [PMID: 38191664; PMID: 39755699; PMID: 40209705]. Have the authors explored any of these more recent approaches to improve cargo (editor and sgRNA) loading in the RENDER system?

Response:

We thank the Reviewer for this question. We have not yet explored the additional engineering strategies they propose for enhancing gene editor packaging into RENDER and we believe that these are excellent suggestions for optimizing a next version of our delivery platform. We agree with the potential of these strategies, particularly improvements such as optimizing Nuclear Export Signal (NES) and Nuclear Localization Signal (NLS) sequences and adding sgRNA-recruiting proteins, which could enhance RENDER’s performance. However, future, more extensive evaluation will be necessary to determine whether these approaches can be applied to RENDER and assess their potential impact.

Additionally, we have recently explored a different engineering strategy referenced in another study [PMID: 39537813], where beneficial capsid mutations for eVLP systems were identified using a directed evolution system. Our preliminary tests with RENDER show that, similar to the reported findings, certain mutations in the viral capsid protein can improve RENDER’s properties. However, since these preliminary results do not alter the key conclusions of our manuscript, and further evaluation is needed to fully assess their effect on RENDER-mediated epigenome editing, we have not included these data in the current version of the manuscript.

Comment 2:

Lines 402–403: The authors should cite prior literature demonstrating that RNP delivery can reduce off-target editing.

Response:

We thank the Reviewer for the suggestion and have added the relevant citations [PMID: 32042165; PMID: 34452911; PMID: 33616436; PMID: 35021064; PMID: 34942274].

Comment 3:

In the iPSC-derived neuron experiment, while authors implied in the text that the plasmid transfection after neuronal differentiation is likely inefficient, it would strengthen the study to include a comparison of plasmid transfection alongside RENDER-mediated delivery in the iPSC-derived neuron as a proper control.

Response:

We thank the Reviewer for the suggestion. Our prior work required delivering CRISPRoff editor plasmid DNA into iPSCs first, sorting for transfected cells, and subsequent differentiation into neurons – a process that is laborious compared to our RENDER platform [PMID: 33838111]. Transient plasmid delivery to neurons remains a significant challenge, particularly compared to iPSCs or other dividing cells, which are amenable to chemical transfection [PMID: 38979269]. Although some commercial reagents for plasmid transfection in neurons are available and there are reports of optimization for this purpose in primary cells [PMID: 30987672; PMID: 40169710], we currently do not have methods for transiently delivering CRISPR editor plasmids into iPSC-derived neurons. We speculate that the size of CRISPR editors may be one factor limiting plasmid delivery success in neurons. However, we and our collaborators have also tested several transfection reagents (e.g. Lipofectamine products) for other types of cargo in iPSC-derived neurons, and they all either fail to deliver or cause massive toxicity, or both. Therefore, we cannot provide plasmid transfection data for comparison with RENDER-mediated delivery in neurons. Nevertheless, as shown in Fig. 5b of our manuscript, the majority of neurons treated with RENDER-CRISPRoff show reduced CD81 expression, indicating that RENDER successfully delivers CRISPRoff to the vast majority of cells, which is necessary for the observed editing. This demonstrates that RENDER delivery outperforms the plasmid transfection efficiency reported in the literature.

Reviewer #2 (Remarks to the Author):

The authors have performed careful and effective revisions. We regret that the TET1 data are not included in this revised version, but we support the acceptance of this work for publication.

Response:

We thank the Reviewer for their positive feedback and support for the acceptance of our work, as well as for their valuable suggestions and the time dedicated to reviewing our manuscript. We also appreciate the Reviewer's understanding.

Reviewer #3 (Remarks to the Author):

Response:

We thank the Reviewer for their positive assessment of our work.

Reviewer #4 (Remarks to the Author):

The authors have addressed all prior concerns satisfactorily. No additional comments/suggestions noted. Good luck to all authors!

Response:

We thank the Reviewer for their positive assessment of our work, as well as for their valuable suggestions and the time spent reviewing our manuscript.